# Philanthropy, audit firms culture and auditor independence

**Yiling Zhang**[1]**, Lang Wei** [2]*

**1** School of Accounting, Southwestern University of Finance and Economics, Chengdu, China, **2** School of Accounting, Chongqing Technology and Business University, Chongqing, China

* weilang1122@163.com

**Data Availability Statement:** The data used in this study is owned by CSMAR (China Stock Market & Accounting Research Database) and CICPA (The Chinese Institute of Certified Public Accountants), therefore, the author has no right to share the data.

## Abstract

In this study, we examine how the prosocial ethical culture in audit firms, measured as their philanthropic contributions over gross revenues, influences auditor independence. Using 5,246 audits in the Chinese market between 2010 and 2012, we find that the level of ethical culture in audit firms is significantly negatively associated with the magnitude of earnings management and the frequencies of financial restatements of their client firms. We also find this association is even stronger when auditors provide services to clients that are economically important or when signing auditors bear school ties with at least one top executive of the client. Further evidence show that the ethical culture can both act as a mechanism that attract auditors with a compatible internal norm, and great a group norm in audit firms that directly shape auditor behavior. Collectively, our study suggests that ethical culture of audit firms can significantly improve auditor independence.

## Introduction

How to promote auditor independence has always been a key question in the auditing professional context, and has a long research history. The existing literature has extensively investigated traditional mechanisms that aim to improve auditor independence, such as enlarging auditors' liability exposure [1, 2], mandatory auditor rotation [3, 4], and restrictions of non-audit services [5, 6], but found mixed evidence regarding their effectiveness. Acknowledging the difficulty to curb accounting scandals and auditing failures, regulators, accounting professions and academic communities have recently drawn more attentions to the potential role of audit firm culture in influencing auditor independence. For example, in the newly published Code of Professional Conduct, AICAP stated that "to eliminate or reduce threat for auditor independence. . .., all audit firms resided in the U.S. should establish a culture of high ethics and integrity" (AICPA, 2014). Additionally, Financial Reporting Council (FRC) in the U.K. also viewed the culture within an audit firm is "a key driver of auditor independence because it has the ability to create an environment that achieving a high level of independence in every aspect of the audit process is valued and rewarded" (FRC, 2006). General Office of the State Council in China stated that "all audit firms should take the construction of integrity as the lifeline of the industry development, always adhere to the development orientation of quality first, and continuously improve the professional ability, independence, moral level and industry credibility of certified public accountants" (GBF, 2021). However, despite the significant

Interested persons can contact CSMAR for the data (see https://www.gtarsc.com/ for more details, contact via 400-639-888) and CICPA (see https://www.cicpa.org.cn/ for more details, contact via 010-8825000). The author confirm that they did not have any special access or privileges to the data that other researchers would not have.

**Funding:** The author(s) received no specific funding for this work.

**Competing interests:** The authors have declared that no competing interests exist.

interests, there is very little empirical evidence regarding the question that whether and how audit firm culture influences auditor independence. In this study, we attempt to answer this question and provide systemic evidence on the role of audit firm culture in influencing auditor independence.

The paucity of empirical evidence on the effect of audit firm culture is primarily because of the difficulty to quantify cultural attributes. As a general matter, audit firm culture is defined as a set of values and beliefs that shared by firms' auditors [7]. Several previous experimental studies try to quantify audit firm culture based on individual auditors' self-report perceptions through using questionnaires, but find a significant variation in these perceptions among different auditors [8–10]. Having assessed the efficacy of some commonly used culture measurements, Key (1999) concludes that measurements based on self-report perceptions are often subjected to a heavy selection problem, and may not accurately reflect cultural characteristics [11]. In addition, since audit firm culture is normally intercepted an organization-level conception, measurements constructed at the individual-level can also have a limited power to capture culture within an audit firm [12]. For better understanding audit firm culture, Jenkins et al. (2008) recommend an investigation level with reasonably homogeneous measurements of audit firm culture [7].

Acknowledging these challenges, we use an audit firm's philanthropic efforts to quantify one aspect of its culture that may affect auditor independence, and focus our investigation at the office-level. The choice of philanthropic efforts as an indicator of audit firm culture is motivated by the observation of recent literature that corporate social responsibility (CSR) activities, such as making philanthropic contributions, often reflect a shared belief and value of committing to social good, that is, a pro-social culture within an organization [13–15]. Meanwhile, given the fact that audit firms are decentralized networks comprised of semi-autonomous individual offices [16, 17], difference offices, even within a same firm, may cultivate a unique and distinct culture. We therefore further construct our culture measurement at the office-level. Specifically, our measurement of the pro-social culture within an audit office is the percentage of philanthropic contributions it has made during a fiscal year over its gross revenue. The promise under this measurement is that when an audit office is more willing to give, it's likely to cultivate a stronger pro-social culture.

An audit office's pro-social culture can help it to establish a positive image of caring for the public interests and refraining from conducting low quality audits [18, 19]. While this image benefits the audit office in various ways [20–22], it also casts additional costs on this office when its auditors are found to compromise independence. Once its auditors' misbehaviors are detected, the negative publicities would not only bring the office with potential harsh sanctions and lawsuits, but also damage its positive image and reduce the associated benefits [23, 24]. Therefore, an audit office which cultivates a stronger pro-social culture is likely to be more motivated to protect its socially responsible image and keep auditors to maintain a high-level independence. In addition, from a perspective of business ethics, pro-social culture is about the moral principles of "doing what is ethically correct" and "to meet the ethical expectations of society", which require an audit office to take account not only economic interests, but also the responsibilities toward clients, communities and society [13, 25, 26]. For ethical concerns, a stronger pro-social culture can also drive an audit office to be less tolerant of auditors' wrongdoings. These arguments suggest that an audit firm's pro-social culture would positively influence auditor independence, consistent with the theoretical prediction that organization culture works as a group norm directly affecting members' behaviors [27–29].

An audit office's pro-social culture can also be positively associated with auditor independence through acting as a selection mechanism that attracts and selects individual auditors who hold similar attitudes toward philanthropic activities to the office [30, 31]. While Auditing

Standards and various office-wide policies left individual auditors limited room to exert idio-syncratic influence on audits, recent literature documents that the differences in auditors' personal characteristics often cause huge heterogeneities in auditing practices [32, 33]. In this sense, the pro-social culture within an audit office can be a manifestation of auditors' personal internal norms to commit to social good. Socially responsible auditors are likely to be sensitive with their images and also have a high level of moral developments, which drive them to behave independently.

We concentrate on the Chinese audit industry since it offers access to information on audit firms as well as the identities of the signing auditors. In addition to the data availability, using the Chinese setting has other advantages. First, the data on audit firms engaging in the philanthropic contribution are nonpublic, which can help us to study auditor's incentives to improve audit quality based on ethical culture, rather than on building a reputation. Second, the other characteristic of the Chinese audit market is of a high degree of dispersion. Wang et al. (2008) find that 54% of Chinese listed firms hired audit offices who are small-scale (non-Top-10) and come from the local site in 2003, whereas only 25% Chinese listed firms hired Top-10 audit offices [34]. Thus, unlike US audit market, Chinese auditors are more likely to succumb to client management power because the client provides a significant proportion of the auditor's income. Third, in China, the market is generally inefficient, and there is weaker legal and enforcement [35]. Due to the greater transaction costs of doing business in this environment, economic players tend to rely more on alternative non-market channels like social ties.

Using Chinese non-financial firms listed in the A-share market between 2010 and 2012 as our sample, we find that 36.3% of sample firm-year are audited by audit firms which proportion of expense of philanthropic contributions on total income exceeds the sample median (0.042). Regression results show that ethical culture of audit firms is significantly negatively associated with the frequencies of financial restatements and the magnitude of earnings management and of their client firms. Further analysis shows that this association is even stronger when auditors provide services to clients that are economically important to or bear a social connection with them. Our results are robust (1) to using alternative measures of audit quality at audit office level, (2) to using alternative designations of ethical culture. In addition, a propensity score matching (PSM) method that takes into account the endogeneity brought on by the choice of auditors produces reliable findings. In order to determine if the relationship between ethical culture and audit quality is causative or merely an association, we also undertake further analyses. According to Van den Steen's (2010) model of corporate culture [36], hiring a new CEO with a different set of beliefs will result in turnover due to both selection and self-sorting. Corporate culture is therefore likely to change significantly around the hiring of a new CEO, despite the fact that it has a tendency to remain over time. In response to this hypothesis, we analyze the impact of ethical culture on audit quality while adjusting for firm fixed effects using a sub-sample of auditor firms with chief accountant changes between 2008 and 2012. We continue to discover a strong positive relationship between ethical culture and audit quality, indicating that time-invariant firm-specific characteristics are not masking the primary findings.

There are hypotheses in the academic literature about how company culture could influence individual behavior [31]. Taking advantage of belief or value measured at the individual level and exploiting respectively data on signing auditors who chose job-hopping between audit firms with high ethical culture and audit firms with low ethical culture in 2010 and 2011, we put these hypotheses to the test as an additional means of addressing endogeneity issues. If the evidence does not support these hypotheses, it is questionable whether the audit firm's ethical culture had any impact on the outcomes.

According to the first channel, an audit firm's ethical culture serves as a selection mechanism by drawing or choosing auditors with similar values or beliefs, and these people

subsequently operate in accordance with those norms, which are then reflected in company profits [30]. Consistent with this channel, we find that audit firms with a high level of ethical culture are more likely to attract auditors with excellent audit quality, and an auditor with low audit quality is more likely to be discharged from them and join those audit firms with a low level of ethical culture.

The second channel predicts that an audit firm's ethical culture can exert its influence outside of internal norms and directly influence auditor behavior through group norms [37]. By controlling for auditor fixed effects and examining the sample of auditors who chose to change jobs during 2010 and 2012, we remove the effect of internal norms to the degree that they are constant. Holding the auditor constant, the results show that the same auditor joins in an audit office with high level of ethical culture, his likelihood of financial restatement and earnings management decrease compare to when he was at an audit office with low level of ethical culture, consistent with ethical culture working through group norms.

Our study contributes to the literature in the following ways. First, we contribute to the literature on the factors that drive auditors to supply audit quality. Prior studies examine reputation and litigation concern, auditor size, auditor industry specialization drive auditors to deliver high audit quality [38–41]. However, there is no literature that ethical culture of the audit firm as a factor in motivating auditors to provide excellent audit quality. We find a positive association between ethical culture of audit firms and their audit quality.

Second, empirical research on corporate culture has been limited in accounting and finance literature mainly due to measure difficulties. In some corporate governance literature, for example, In Parsons et al. (2018), local geographic culture measures are used around the company's headquarters [42]; Bereskin et al. (2013) find that corporate foundation giving reflects a company's culture [43]. In auditing literature, Bik and Hooghiemstra (2017) take culture into account but only examine the effect of national culture on audit quality [44]. Complementing these earlier studies, this paper measures the ethical culture of audit firms using the size-adjusted anonymous philanthropic contributions over the total income of each auditor office in China.

Third, we provide evidence on the inner workings of ethical culture, which shows that the culture influences auditor behavior in two ways, one is acting as an auditor selection mechanism for the audit offices and another one is directly influencing the auditor's individual behavior.

## Literature review and hypothesis development

Culture, broadly defined, refers to "traditions, customs, moral values, religious beliefs, and all other forms of behavior that have passed the test of time" [45]. Human behavior is normalized through these unwritten rules because they provide society members with schematic, mental models of what is "good" versus "bad", what is "appropriated" versus "inappropriate", and they set norms for behavior [46]. According to this idea, auditors, as members of society, are not only more likely to be aware of social rules, but they are also more likely to expect that others will follow those rules as well. In other words, auditors' behavior, especially for their professional judgments, are likely to be affected by the schematic, mental model prevailing in a society [47, 48]. Prior research finds cultural differences in auditors' compliance with company policies affect their assessment procedures for fraud risk [49]. Specifically, they find that collectivism and societal trust are negatively associated with compliance with global firm policy, while religiosity is positively associated with compliance. Nolder and Riley (2014) review the audit literature that the association culture with variation in auditor's ethical judgment [50]. And it is proposed that researchers should pay attention to the effect of audit firm culture on auditor's judgment and decision-making.

There is growing literature that examines the effect of auditor's characteristics on audit quality [22]. However, there has been little study of the role of audit firm culture and ethics on audit quality. The reason for this is partly due to the difficulty of developing an empirical measure of audit firm culture and ethics.

Research on corporate governance addresses this difficulty by measuring firm culture based on corporate philanthropy [43]. According to Schein (1985), "artifacts" are visible manifestations of an organization's underlying values [51]. To the extent that philanthropy can be seen as an artifact of a firm's culture, it can reflect socially responsible values and influence the propensity for wrongdoing.

Corporate giving has been extensively studied in a wide variety of disciplinary contexts. The primary reason for corporate giving is that it can enhance a firm's public image and generate a positive image of the company. Fisman et al. (2006) models corporate philanthropy as a signal of trust where product quality is intangible [52]. In addition, a firm's corporate giving is likely to be associated with a culture of high regard for its reputation if it is motivated by a desire for high integrity and building trust with customers. Therefore, these firms doing giving are reluctant to engage in misconduct since they spend a lot of time and money building their image, and engaging in misconduct comes at a higher price [53, 54]. Therefore, audit quality is the output product for the audit firms, when stakeholders, such as regulators, are unable to judge audit quality, the audit firm's giving can serve as a signal of a high-audit quality. Moreover, since donation audit firms have spent a great deal of time and energy building reputations, they will be more adamant about maintaining professional independence to prevent reputation losses caused by audit failures.

In addition to the above literature, there are also views that corporate giving as a manifestation of corporate social responsibility stems from an altruistic or pro-social behavior [55]. Brown and Ferris (2004) stated that "selfless or not, these acts involve a degree of compassion and commitment to others [56]". The auditor's prosocial ethics may influence his behavior and reduce his willingness to tolerate clients' opportunistic behavior.

Such a role for personal ethics in the firm's propensity to engage in wrongdoing is also supported by recent papers. Biggerstaff et al. (2012) find that Executives who are unethical are more likely to manipulate earnings [57]. Davidson et al. (2012) track unethical behavior based on past legal infractions, like driving under the influence [58]. Using this measure, they find that executive misconduct is more likely to occur. If corporate giving is a manifestation of the auditor's pro-social ethic, it is likely to increase their propensity to insist on professional independence. This leads us to our first hypothesis,

H1: Ethical culture in audit firms, measured as their philanthropic contributions over total incomes, is positively associated with audit quality.

Due to large start-up costs associated with each individual client, incumbent auditors enjoy a cost advantage over rivals and generate quasi rents (i.e., audit fees that are higher than audit costs) in successive audits. As a result, incumbent auditors are not expected to be totally independent of their clients because the latter could incur significant costs for the former by severing the relationship [39]. Generally speaking, the larger the customer in an auditor's portfolio, the greater the motivation the auditor has to keep the client, thereby lowering the audit quality.

Because of significant client-specific start-up costs, incumbent auditors have cost advantages over competitors and earn quasi rents (ie., audit fees in excess of audit costs) in subsequent audits. Hence, incumbent auditors are not expected to be completely independent of their clients, as the latter can impose real cost on the former by terminating the bilateral relationship [39]. Ceteris paribus, the larger client in an auditor's protifolio, the incentive that the auditor has to retain the client and thus possibly compromise audit quality. Chen et al. (2010)

find that when the mechanisms for investor protection are inadequate in China, auditors are more inclined to compromise audit quality for clients who are economically significant [59]. Therefore, we should see a stronger association between ethical culture of audit firms and its audit quality for audit of those important clients. We investigate this question in the following hypothesis:

H2: The positive association between ethical culture of audit firms and audit quality is stronger (more positive) for audit of more important clients than for audit of less important clients.

One type of social links that may have an impact on the decision-making process is one that results from sharing an educational connection. Guan et al. (2016) suggest that compared to the links made through former employment investigated in earlier studies, school relationships can be a better proxy for social connections to test their impact on decisions made in the audit scenario [60]. Because school ties can make clear judgments regarding the influence of social connections on audit outcomes [60]. Furthermore, managers have a strong propensity to collude with auditors because they serve as the controlling shareholders' representatives. Under pressure from managers, auditors are likely to sacrifice their independence given the intense competition they face in retaining existing clients and acquiring new ones. Therefore, Guan et al. (2016) show that when there are school ties between auditors and executives, there is a higher likelihood of obtaining a favorable audit opinion, as well as higher reported discretionary accruals and audit fees, than when there are none [60]. In this paper, we examine the moderating effect of school ties at audit office level, we predict that if there is a school tie between the signing auditors and the top executives of the client company, the positive association between ethical culture and audit quality is stronger. We test this prediction in the following hypothesis:

H3: The positive association between ethical culture and audit quality is stronger when signing auditors bear school ties with at least one top executives of the client.

## Model specification and data

### Sample and data

From the China Stock Market and Accounting Research (CSMAR) database, we start with the population of Chinese non-financial enterprises listed in the A-share between 2010 and 2012. The data on financial restatements from Chinese Research Data Services Platform. The proprietary dataset on philanthropic contributions of audit offices was got from Chinese Institute of Certified Public Accountants (CICPA), which is unpublic. The first sample consists of 5,749 firm-year observations of listed enterprises. Because these companies apply International Accounting Standards in addition to Chinese Generally Accepted Accounting Principles (GAAP) and have an external auditor, we remove 50 observations of listed companies that issue B shares [1]. We then exclude 130 companies that are finance sectors, because these companies in this industry have different reporting regulations. In addition, we exclude 47 observations of listed companies without information whether the expenditure of audit offices controlled by its audit firm. Furthermore, 276 observations of listed companies with insufficient financial information are disregarded. Finally, our sample consists of 5246 audits of listed companies. The sample collecting process is displayed in Panel A of Table 1.

In compliance with Chinese auditing requirements, Chinese auditors are required to sign audit reports to identify who is responsible for the audit (Ministry of Finance, 1995). We thus manually collect the identities of audit office and signing auditors from annual reports. Data on audit offices' location are obtained from the enquiry system compiled by the CICPA at http://cmis.cicpa.org.cn, and the websites of audit firms. To determine whether shareholders of audit offices have changes, we checked all the relevant information from websites of audit

**Table 1. Sample selection process.**

| Panel A: Sample collection | |
|---|---|
| | Total firm-year observations |
| Total A-share listed companies from 2010 to 2012 | 5,749 |
| Less: B-share companies | (50) |
| Less: Financial companies | (130) |
| Less: companies without information whether the expenditure of audit offices controlled by its audit firm | (47) |
| Less: companies that have incomplete financial information | (276) |
| Final sample | 5,246 |

| Panel B: sample client companies | | | |
|---|---|---|---|
| | Number (percentage) of observations | | |
| | Audited by audit offices with high ethical culture | Audited by audit offices with low ethical culture | Total |
| (1)Total number | 1,903(36.3%) | 3,343 (63.7%) | 5,246 |
| (2)Number of client companies | | | |
| Year 2010 | 455 (41.1%) | 652 (58.9%) | 1,107 |
| Year 2011 | 592 (30.0%) | 1,381 (70.0%) | 1,973 |
| Year2012 | 856 (39.5%) | 1,310 (60.5%) | 2,166 |

| Panel C: Sample audit offices | | | |
|---|---|---|---|
| | Number (percentage) of observations | | |
| | Audit offices with high ethical culture | Audit offices with low ethical culture | Total |
| Number of audit offices | | | |
| Year 2010 | 32 (40.5%) | 47(59.5%) | 79 |
| Year 2011 | 44(20.3%) | 173(79.7%) | 217 |
| Year2012 | 52(22.5%) | 179(77.5%) | 231 |

firms. If anyone shareholders of audit offices changed between 2008 and 2012, we divide them into subsample of appointment of new chief auditors.

The summary statistics for the client firms and audit offices are shown in Panels B and C of Table 1, respectively. As shown in Panel B, 36.3 percent (1903 audits) of observations are audited by audit offices with high ethical culture and 63.7 percent (3343 audits) of observations are audited by audit offices with low ethical culture. The number and percentage of audits that are conducted by audit office with high ethical culture increase during the same period. As reported in Panel C, there are no significant difference between the proportion of audit offices with high ethical level and the proportion of audit offices with low ethical level.

## Research design

To test our H1 that whether there is positive association between ethical culture in audit firms and their audit quality, we estimate the following model of audit quality:

$$
\begin{aligned}
RE_{i,t} = {} & \alpha_0 + \alpha_1 PHIL_{i,t} + \alpha_2 ASSET_{i,t} + \alpha_3 LEV_{i,t} + \alpha_4 ROE_{i,t} + \alpha_5 LOSS_{i,t} + \alpha_6 CURRENT_{i,t} \\
& + \alpha_7 AR_{i,t} + \alpha_8 INV_{i,t} + \alpha_9 BIG4_{i,t} + \alpha_{10} LOCAL_{i,t} + \alpha_{11} CLIENT_{i,t} + \alpha_{12} AGE_{i,t} \\
& + \alpha_{13} SOE_{i,t} + \alpha_{14} SCHOOLTIE_{i,t} + \sum \delta YearDummy_{i,t} + \sum \gamma IndustryDummy_{i,t} \\
& + \varepsilon_{i,t}
\end{aligned}
\tag{1}
$$

$$DA_{i,t} = \alpha_0 + \alpha_1 PHIL_{i,t} + \alpha_2 ASSET_{i,t} + \alpha_3 LEV_{i,t} + \alpha_4 ROE_{i,t} + \alpha_5 LOSS_{i,t} + \alpha_6 CURRENT_{i,t}$$
$$+ \alpha_7 GROWTH + \alpha_8 BIG4_{i,t} + \alpha_9 LOCAL_{i,t} + \alpha_{10} CLIENT_{i,t} + \alpha_{11} AGE_{i,t}$$
$$+ \alpha_{12} SOE_{i,t} + \alpha_{13} SCHOOLTIE_{i,t} + \alpha_{14} INDDIR_{i,t} + \alpha_{15} DUAL$$
$$+ \sum \delta YearDummy_{i,t} + \sum \gamma IndustryDummy_{i,t} + \varepsilon_{i,t} \qquad (2)$$

The dependent variable, we use financial restatement and accruals to proxy for audit quality. First, in Eq (1), the dependent variable *RE*, financial restatement, clearly indicates that audit quality can be measured by whether or not the auditor issued an unqualified opinion in the case of materially misstated financial statements [61]. *RE* is a dummy variable and has a value of one if a client company restates in the current fiscal year, otherwise it has a value of zero. Second, in Eq (2), the dependent variable *DA*, accrual, we employ performance-adjusted discretionary accruals as proxy variables for audit quality using the model suggested by Kothari et al. (2005) [62], which is computed as follow. For each two-digit SIC code industry and year with a minimum of 10 observations, we estimate the cross-sectional version of the modified Jones model in Eq (3). Residuals from Eq (3) are *DA* before adjusting for firm performance.

$$ACCR_{jt}/TA_{jt-1} = \alpha_1 [1/TA_{jt-1}] + \alpha_2 [(\Delta REV_{ij} - \Delta REC_{ij})/TA_{jt-1}] + [PPE_{jt}/TA_{jt-1}] + \varepsilon \quad (3)$$

where $ACCR_{jt}$ is total accruals in year t; $TA_{jt-1}$ is total asset in year t-1; $\Delta REV_{ij}$ is changes in net sales in year t; $\Delta REC_{ij}$ is changes in receivables in year t; $PPE_{jt}$ is gross property, plant and equipment in year t. The discretionary accruals denoted as *DA* represent the difference between total accruals and the estimated (fitted) normal accruals. The higher the absolute value of discretionary accruals, the lower the audit quality.

Our variable of interest, $PHIL_{i,t}$, equals the ratio of the expense of the philanthropic contribution of auditor office *i* in *t* year on the total income of the auditor office *i* in *t* year if the expense of auditor office *i* is not governed by its audit firm. However, when the expense of auditor office *i* is governed by its audit firm, $PHIL_{i,t}$ is calculated as the sum of the expense of philanthropic contribution of all the auditor office of its audit firm and the expense of philanthropic contribution of its audit firm itself, scaled by the sum of total income of all the auditor office of its audit firm and the total income of its audit firm itself in year *t*. Therefore, according to hypothesis, we predict that $PHIL_{i,t}$ will be negatively associated with both *RE* in Eq (1) and in Eq (2).

For control variables in Eq (1), we include client company financial characteristics such as leverage (*LEV*), the presence of loss (*LOSS*), size (*ASSET*), profitability (*ROE*), the current ratio (*CURRENT*), and listing age (*AGE*), because these characteristics are likely to influence audit quality [2, 34, 63]. According to previous research in China, the listed companies' risks are influenced by state ownership [34, 64]. Due to these findings, we include a variable indicating whether a firm is state-owned (*SOE*).

In addition, we control for a number of auditor characteristics. Several studies have found that large audit firms provide high-quality services [39, 65, 66]. To control for the effects of audit firm size on audit quality, we include a variable (*BIG4*) and assign it to one of a company hires one of Big Four international audit firms, and zero otherwise. Wang et al. (2008) show that client companies in the same region are more likely to hire auditors in China [34]. Accordingly, we include a variable (*LOCAL*) and assign it to one if listed companies hire non-big4 audit firms' auditors who work in the same city as the listed companies. We additionally control for client importance (*CLIENT*) at the audit office level since the relative importance

of a client to an auditor can affect audit quality [59]. Besides, Guan et al. (2016) suggest that it is more likely that companies audited by connected auditors report higher discretionary accruals, and to restate earnings downward in the future, we include a variable (*SCHOOLTIE*) that equal one if either of the client company's top executives attended the same university as either of its signing auditors, and zero otherwise [60].

When the dependent variable in Eq (1) is financial restatements, we further control for account receivables (*AR*), inventory (*INV*) because these characteristics are related to audit risk and complexity of clients.

In Eq (2), we also include client company financial characteristics such as size (*ASSET*), leverage (*LEV*), the presence of loss (*LOSS*), profitability (*ROE*), the current ratio (*CURRENT*), and listing age (*AGE*), audit firm size (*BIG*4), the firm whether are ultimately controlled by the government(*SOE*), whether is non-BIG4 and auditor office is located in the same city as the client (*LOCAL*), school ties (*SCHOOLTIE*) and client's importance (*CLIENT*). In addition, we control the growth rate of total sales (*GROWTH*), and to represent the corporate governance mechanisms, we also include *INDDIR*, which is the ratio of independent directors on boards of client companies and *DUAL*, which is a dummy to indicate whether the client's chairman and CEO are the same. Finally, to account for any differences in audit quality between industries and across time, we incorporate fixed effect dummies for industries and years in Eqs (1) and (2).

To test hypothesis H2, which examines how the significance of clients affects financial restatements and ethical culture in audit firms, we modify Eq (1) and Eq (2) by estimating the following regression model:

$$
\begin{aligned}
RE_{i,t} = {} & \alpha_0 + \alpha_1 PHIL_{i,t} + \alpha_2 PHIL_{i,t} * client_{i,t} + \alpha_3 ASSET_{i,t} + \alpha_4 LEV_{i,t} + \alpha_5 ROE_{i,t} \\
& + \alpha_6 LOSS_{i,t} + \alpha_7 CURRENT_{i,t} + \alpha_8 AR_{i,t} + \alpha_9 INV_{i,t} + \alpha_{10} BIG4_{i,t} + \alpha_{11} LOCAL_{i,t} \\
& + \alpha_{12} CLIENT_{i,t} + \alpha_{13} AGE_{i,t} + \alpha_{14} SOE_{i,t} + \alpha_{15} SCHOOLTIE_{i,t} \\
& + \sum \delta YearDummy_{i,t} + \sum \gamma IndustryDummy_{i,t} + \varepsilon_{i,t}
\end{aligned}
\tag{4}
$$

$$
\begin{aligned}
|DA|_{i,t} = {} & \alpha_0 + \alpha_1 PHIL_{i,t} + \alpha_2 PHIL_{i,t} * client_{i,t} + \alpha_3 ASSET_{i,t} + \alpha_4 LEV_{i,t} + \alpha_5 ROE_{i,t} \\
& + \alpha_6 LOSS_{i,t} + \alpha_7 CURRENT_{i,t} + \alpha_8 GROWTH + \alpha_9 BIG4_{i,t} + \alpha_{10} LOCAL_{i,t} \\
& + \alpha_{11} CLIENT_{i,t} + \alpha_{12} AGE_{i,t} + \alpha_{13} SOE_{i,t} + \alpha_{14} SCHOOLTIE_{i,t} + \alpha_{15} INDDIR_{i,t} \\
& + \alpha_{16} DUAL + \sum \delta YearDummy_{i,t} + \sum \gamma IndustryDummy_{i,t} + \varepsilon_{i,t}
\end{aligned}
\tag{5}
$$

To facilitate interpreting the results, we use the natural logarithm of a client's total assets as a surrogate to measure client economic importance [59]. This is suitable because audit fee in China is typically based in total client assets, and the logarithm transformation takes the non-linear relationship between audit fee and client total assets into account. We define the importance of client *i* to auditor office *j* as:

$$
CLIENT = \frac{Ln\ ASSET}{\sum_{i=1}^{n} LnASSET}
\tag{6}
$$

Where $LnASSET_i$ is the natural logarithm of the total assets of client *i* and $\sum_{i=1}^{n} LnASSET_i$ is the sum of the total assets (in natural logarithm form) of *n* clients audited by auditor office *j* in a particular year. In addition, we set *client* to 1 of the importance of client *i* in yeat *t* is greater than the sample mean for auditor office *j*, and 0 otherwise. Coefficient $\alpha_2$ reflects the incremental probability of financial restatements when audit offices audit more important clients as

compared with less important clients. Based on the hypothesis 2, we expect $\alpha_1 < 0$ and $\alpha_2 < 0$ in Eq, (4) and Eq (5). The control variables are the same as in Eq (1) and Eq (2).

To test H3, which focuses on the moderating effect of school ties on the influence of ethical culture, we supplement Eq (1) and Eq (2) with an interaction term and estimate the following regression model:

$$RE_{i,t} = \alpha_0 + \alpha_1 PHIL_{i,t} + \alpha_2 PHIL_{i,t}*SCHOOLTIE_{i,t} + \alpha_3 ASSET_{i,t} + \alpha_4 LEV_{i,t} + \alpha_5 ROE_{i,t}$$
$$+ \alpha_6 LOSS_{i,t} + \alpha_7 CURRENT_{i,t} + \alpha_8 AR_{i,t} + \alpha_9 INV_{i,t} + \alpha_{10} BIG4_{i,t} + \alpha_{11} LOCAL_{i,t}$$
$$+ \alpha_{12} CLIENT_{i,t} + \alpha_{13} AGE_{i,t} + \alpha_{14} SOE_{i,t} + \alpha_{15} SCHOOLTIE_{i,t}$$
$$+ \sum \delta YearDummy_{i,t} + \sum \gamma IndustryDummy_{i,t} + \varepsilon_{i,t} \qquad (7)$$

$$|DA|_{i,t} = \alpha_0 + \alpha_1 PHIL_{i,t} + \alpha_2 PHIL_{i,t}*SCHOOLTIE_{i,t} + \alpha_3 ASSET_{i,t} + \alpha_4 LEV_{i,t} + \alpha_5 ROE_{i,t}$$
$$+ \alpha_6 LOSS_{i,t} + \alpha_7 CURRENT_{i,t} + \alpha_8 GROWTH + \alpha_9 BIG4_{i,t} + \alpha_{10} LOCAL_{i,t}$$
$$+ \alpha_{11} CLIENT_{i,t} + \alpha_{12} AGE_{i,t} + \alpha_{13} SOE_{i,t} + \alpha_{14} SCHOOLTIE_{i,t} + \alpha_{15} INDDIR_{i,t}$$
$$+ \alpha_{16} DUAL + \sum \delta YearDummy_{i,t} + \sum \gamma IndustryDummy_{i,t} + \varepsilon_{i,t} \qquad (8)$$

where $SCHOOLTIES_{i,t}$ equal one if any of the client's top executives attended the same alma mater as the signing auditors, and zero otherwise. Specifically, regardless of whether they attended the same schools at the same times, on the same campuses, or for the same major, this variable indicates whether they attended the same universities for undergraduate or graduate degrees. The coefficient for the interaction term $PHIL_{i,t}*SCHOOLTIES_{i,t}$ captures the incremental influence of school ties between signing auditors and the client company's top executives. Based on H3, we expect $\alpha_2 < 0$ in Eq (7) and Eq (8).

The Table 2 contains detailed variable definitions. Since the key variable of interest, $PHIL_{i,t}$, varies by firm and year, standard errors are clustered by firm so that we can account for residual correlations within firms.

## Empirical results and analysis

### Descriptive statistics

The descriptive statistics for all the variables utilized in this study are shown in Table 3. As Table 3 reports, the percentage of firm-year observations that occurred financial restatements is 15.6% over the sample period, which is comparable to that reported by Wang et al. (2015) [33]. The average magnitudes of $|DA|$ is 0.083, which is very similar to previous studies [3, 67].

Of all the sample observations, we find 36.3% the expense of philanthropic contributions over total income of our sample observations beyond 0.055. Only 5.4% of the research sample's companies employ Big Four auditors. In addition, the sample with school ties only account for 1.7% of the research sample. Moreover, the mean of client importance at auditor office level is 9.6%, which is also similar to what is reported in previous studies [59].

### Benchmark regression

In this section, we conduct regression analyses to test our first hypothesis that philanthropic auditors are more likely to improve audit quality. The main regression results for financial restatements, earning management are presented in column (1) and column (2) of Table 4. Interestingly, in two regressions of ethical culture, both coefficients are negative and statistically significant (p<0.01), indicating that the ethical culture of audit firms is positively associated with the quality of their audits, which is consistent with the main prediction.

**Table 2. Variable definitions.**

| Variables | Definition |
|---|---|
| RE | Indicator variable that equals 1 if client companies is subsequently restate in the current fiscal year, and zero otherwise. |
| \|DA\| | We use a cross-sectional version of modified Jones model to estimate discretionary accruals. |
| MAO | Indicator variable that equals 1 if client companies received MAOs in the current fiscal year, and zero otherwise. |
| PHIL | The ratio of the expense of the philanthropic contribution of auditor office in current fiscal year on the total income of the auditor office in t year. |
| ASSET | Natural logarithm of the total assets of the client company. |
| LEV | The ratio of the total liability to total assets of the client company. |
| ROE | The ratio of core operating net income to the ending shareholder equity. |
| LOSS | Indicator variable that equals 1 if the client experiences losses in the current fiscal year, and 0 otherwise. |
| CURRENT | The ratio of current assets to current liabilities at the end of the year. |
| GROWTH | The growth rate of the client company's total sales. |
| AR | The percentage of accounting receivables over total assets of a client company. |
| INV | The percentage of inventory over total assets of a client company. |
| BIG4 | Indicator variable that equals 1 if the auditor is one of the Big Four CPA firms, and 0 otherwise. |
| LOCAL | Indicator variable that equals 1 if the auditor is from a local firm, and 0 otherwise. |
| CLIENT | Indicator variable that equals 1 if the importance of client $i$ in yeat $t$ is greater than the sample mean for auditor office $j$, and 0 otherwise. |
| AGE | The number of years that the company has been listed on the stock market. |
| SOE | Indicator variable that equals 1 if the client is ultimately controlled by central or local governments, and 0 otherwise. |
| SCHOOLTIE | Indicator variable that equals 1 if any of the client company's top executives has a common alma mater with either of the signing auditors, and zero otherwise. |
| DUAL | Indicator variable that equals 1 if chairman and CEO are the same person, and 0 otherwise. |
| INDDIR | The percentage of the number of independence directors in board. |
| HOPPING1 | Indicator variable that equals to one if auditors chose job-hopping from audit offices with low ethical culture to audit offices with high ethical culture in given year, and 0 otherwise. |
| HOPPING2 | Indicator variable that equals to one if auditors chose job-hopping from audit offices with low ethical culture to audit offices with low ethical culture in given year, and 0 otherwise. |
| HOPPING3 | Indicator variable that equals one if any one of signing auditors who chose job-hopping from audit offices with high ethical culture to audit offices with low ethical culture in a given year, and 0 otherwise. |
| HOPPING4 | Indicator variable that equal to one if auditor chose job-hopping audit offices with high ethical culture to audit offices with high ethical culture in a given year, and 0 otherwise |

In column 1, the dependent variable is financial restatements, which is a dummy that equals one (zero otherwise) if the annual report of client $i$ is subsequently restated in year $t$. The coefficient on ethical culture is -2.043 ($t = -2.755$). In terms of economic effect, a one standard deviation (0.905) increase in the audit office's ethical culture is associated with a decrease in the presence of accounting error of 1.849%, which is 11.9% of the mean financial restatements rate of 15.6%.

In column 2, the dependent variable is discretionary accruals. The coefficient on ethical culture is -0.084 (t = -3.654). In terms of economic effects, a one standard deviation (0.905) increase in the audit office's ethical culture is associated with a decrease in earnings management of 0.076%, which is 0.92% of mean discretionary accruals of 8.3%.

Column (3) and column (4) of Table 4 show the estimation results for the moderating effects of the client importance (H2). As expected, in column (3) and column (4), the coefficients for *PHIL* is -2.012(-0.058) and is significant at p<0.05 (p<0.005), and the coefficients

**Table 3. Descriptive statistics.**

| Variable | N | Mean | S.D. | Min | p25 | p50 | p75 | Max |
|----------|---|------|------|-----|-----|-----|-----|-----|
| RE | 5,246 | 0.156 | 0.363 | 0 | 0 | 0 | 0 | 1 |
| \|DA\| | 4,692 | 0.083 | 0.102 | 0 | 0.023 | 0.052 | 0.101 | 0.617 |
| PHIL | 5,246 | 0.055 | 0.056 | 0 | 0.012 | 0.042 | 0.074 | 0.236 |
| ASSET | 5,246 | 21.77 | 1.264 | 19.24 | 20.86 | 21.58 | 22.49 | 25.78 |
| LEV | 5,246 | 0.067 | 0.102 | 0 | 0 | 0.011 | 0.100 | 0.464 |
| ROE | 5,246 | 0.076 | 0.106 | -0.478 | 0.039 | 0.076 | 0.116 | 0.410 |
| LOSS | 5,246 | 0.077 | 0.266 | 0 | 0 | 0 | 0 | 1 |
| CURRENT | 5,246 | 3.195 | 4.516 | 0.241 | 1.077 | 1.653 | 3.155 | 29.70 |
| GROWTH | 4,613 | 0.227 | 0.640 | -0.698 | -0.009 | 0.137 | 0.301 | 4.971 |
| AR | 5,246 | 0.099 | 0.092 | 0 | 0.024 | 0.076 | 0.147 | 0.410 |
| INV | 5,246 | 0.166 | 0.154 | 0 | 0.066 | 0.128 | 0.208 | 0.756 |
| BIG4 | 5,246 | 0.054 | 0.225 | 0 | 0 | 0 | 0 | 1 |
| LOCAL | 5,246 | 0.278 | 0.448 | 0 | 0 | 0 | 1 | 1 |
| CLIENT | 5,246 | 0.096 | 0.172 | 0.006 | 0.019 | 0.040 | 0.091 | 1 |
| AGE | 5,246 | 8.197 | 6.273 | 0 | 2 | 8 | 14 | 20 |
| SOE | 5,246 | 0.446 | 0.497 | 0 | 0 | 0 | 1 | 1 |
| SCHOOLTIE | 5,246 | 0.017 | 0.128 | 0 | 0 | 0 | 0 | 1 |
| DUAL | 5,246 | 0.241 | 0.427 | 0 | 0 | 0 | 0 | 1 |
| INDDIR | 5,246 | 0.369 | 0.052 | 0.308 | 0.333 | 0.333 | 0.400 | 0.571 |

for *PHIL\*CLIENT* is -6.154(-0.114) and is significant at p<0.01(p<0.05), indicating the association between ethical culture and audit quality is stronger for audit of more important clients than for audit of less important clients. Overall, these results support the interpretation that economic incentives amply the impact of ethical culture in audit firms on audit quality.

Column (5) and column (6) of Table 4 report the estimation results for the moderating effects of school ties (H3). As expected, in column (5), the coefficient for *PHIL* and *PHIL\*SCHOOLTIES* are -1.989 (p<0.01) and -0.082 (p<0.01), respectively, suggesting that the influence of ethical culture on reducing financial restatements is greater for audit of client which at least one top executives of client companies bear school ties with any one of signing auditors. In column (6), the coefficient for *PHIL\*SCHOOLTIES* are -12.004 (p<0.05) and -0.000 (p<0.01), respectively, suggesting the impact of ethical culture on discretionary accruals is more pronounced when at least one top executives of client companies bear school ties with any one of signing auditors. Therefore, we fail to reject the null hypothesis of H3.

## Robustness

1. Robustness Test I: Alternative measure of audit quality. We use the auditor's tendency to issue a modified audit opinion (MAO) [1, 59, 61] as another alternative surrogate for audit quality at the audit office level. We augment Eq (1) with firm fixed effects. In Column (1) of Table 5 shows that the coefficient for $PHIL_{i,t}$ is 3.984 (p< 0.05). This result using alternative measures of audit quality reinforces our main inference.

2. Robustness Test II: Alternative measures of ethical culture. Foundation giving over the entire sample period is examined as a proxy for ethical culture. We calculated the mean and median foundation giving per audit office for the entire sample period from 2010 to 2012. Robustness tests were performed using mean and median as surrogate variables for ethical culture and defined as *phil*. The results are shown in columns (2) to column (5) of Table 5. The

**Table 4. Benchmark regression.**

| | Model1 | Model2 | Model3 | Model4 | Model5 | Model6 |
|---|---|---|---|---|---|---|
| | *RE* | *\|DA\|* | *RE* | *\|DA\|* | *RE* | *\|DA\|* |
| | **(1)** | **(2)** | **(3)** | **(4)** | **(5)** | **(6)** |
| PHIL | -2.043*** | -0.084*** | -2.012** | -0.058** | -1.989*** | -0.082*** |
| | (-2.755) | (-3.654) | (-2.367) | (-2.011) | (-2.661) | (-3.537) |
| PHIL*CLIENT | | | -6.154*** | -0.114** | | |
| | | | (-3.364) | (-2.560) | | |
| PHIL*SCHOOLTIE | | | | | -12.004** | -0.000*** |
| | | | | | (-1.994) | (-3.047) |
| ASSET | -0.153*** | -0.007*** | -0.140*** | -0.007*** | -0.151*** | -0.007*** |
| | (-3.332) | (-4.627) | (-2.994) | (-4.670) | (-3.278) | (-4.632) |
| LEV | 0.589 | -0.010 | 0.492 | -0.001 | 0.592 | -0.010 |
| | (1.127) | (-0.654) | (0.926) | (-0.072) | (1.134) | (-0.652) |
| ROE | -0.191 | 0.041 | -0.087 | 0.059** | -0.150 | 0.041 |
| | (-0.428) | (1.642) | (-0.194) | (2.383) | (-0.344) | (1.639) |
| LOSS | 0.505*** | 0.040*** | 0.563*** | 0.039*** | 0.521*** | 0.040*** |
| | (3.130) | (5.156) | (3.476) | (5.092) | (3.245) | (5.149) |
| CURRENT | -0.015 | -0.002*** | -0.014 | -0.002*** | -0.014 | -0.002*** |
| | (-1.369) | (-4.761) | (-1.305) | (-5.178) | (-1.327) | (-4.772) |
| AR | -0.844 | | -0.888 | | -0.827 | |
| | (-1.548) | | (-1.610) | | (-1.517) | |
| INV | -0.401 | | -0.356 | | -0.426 | |
| | (-1.185) | | (-1.050) | | (-1.254) | |
| BIG4 | -0.949*** | -0.002 | -0.946*** | -0.004 | -0.952*** | -0.002 |
| | (-3.205) | (-0.340) | (-3.195) | (-0.680) | (-3.215) | (-0.335) |
| LOCAL | 0.048 | -0.000 | 0.074 | 0.000 | 0.051 | -0.000 |
| | (0.512) | (-0.079) | (0.775) | (0.029) | (0.539) | (-0.073) |
| CLIENT | 0.186 | -0.013** | 0.274** | 0.002 | 0.195 | -0.012* |
| | (0.815) | (-2.121) | (2.403) | (0.369) | (0.855) | (-1.953) |
| AGE | 0.012 | 0.001*** | 0.009 | 0.001*** | 0.011 | 0.001*** |
| | (1.389) | (2.734) | (1.070) | (2.589) | (1.341) | (2.716) |
| SOE | 0.012 | 0.001*** | -0.164 | -0.009** | -0.148 | -0.008** |
| | (1.389) | (2.734) | (-1.596) | (-2.478) | (-1.454) | (-2.452) |
| SCHOOLTIE | -0.150 | -0.008** | -0.084 | -0.008 | 0.097 | -0.007 |
| | (-1.481) | (-2.453) | (-0.267) | (-0.735) | (0.235) | (-0.675) |
| GROWTH | | 0.017*** | | -0.000** | | 0.017*** |
| | | (3.886) | | (-2.292) | | (3.884) |
| DUAL | | -0.000 | | -0.000 | | -0.000 |
| | | (-0.115) | | (-0.078) | | (-0.129) |
| INDDIR | | 0.022 | | 0.023 | | 0.022 |
| | | (0.822) | | (0.868) | | (0.810) |
| _cons | 1.304 | 0.209*** | 1.004 | 0.218*** | 1.240 | 0.209*** |
| | (1.238) | (6.302) | (0.938) | (6.443) | (1.178) | (6.307) |
| Year Dummy | Include | Include | Include | Include | Include | Include |
| Industry Dummy | Include | Include | Include | Include | Include | Include |
| N | 5,246 | 4,613 | 5,246 | 4,613 | 5,246 | 4,613 |

(*Continued*)

**Table 4.** (Continued)

|  | Model1 | Model2 | Model3 | Model4 | Model5 | Model6 |
|---|---|---|---|---|---|---|
|  | *RE* | *\|DA\|* | *RE* | *\|DA\|* | *RE* | *\|DA\|* |
|  | (1) | (2) | (3) | (4) | (5) | (6) |
| Pseudo R²/adj. R² | 2.7% | 27.5% | 3.0% | 26.5% | 2.7% | 27.5% |

Note

*, ** and *** indicate that the regression coefficients are significant at confidence levels of 10%,5% and 1%, respectively. *RE*, restate; *DA*, discretionary accruals; *PHIL*, philanthropic contribution of auditor office.

coefficients on the ethical culture variable are significantly negative, which once again confirms the robustness of our results.

3. Robustness Test III: Address endogeneity problems. The last section shows that ethical culture is negatively associated with the frequencies of financial restatements and the magnitude of earning management and these relations hold after controlling for year fixed effect and industry fixed effect. In order to examine whether the correlations are causal or merely associations, we conduct several additional analyses. First, we examine audit quality around the time of new chief auditor appointments while controlling for time-invariant firm characteristics through audit office fixed effect. Second, there are potential endogenous between ethical culture in audit firms and their client–i.e., audit offices are more likely to select client with great performance according to own characteristics. We use propensity score matching to address these concerns.

Van den Steen (2010) devises a model of corporate culture and predicts that the appointment of a new CEO will lead to turnover through both selection and self-sorting [36]. Consistent with this prediction, Hayes et al. (2006) show that the likelihood of top management turnover increases markedly around times of CEO turnover [68]. We also find support for this prediction in this section. The chief aditor of the audit office is the main leader in charge of the office and is responsible for planning and managing the firm's various audit engagements and teams. Trevino et al. (1998) find that ethical culture can be regarded as a subset of organizational culture that represents the multidimensional interplay among formal and informal systems of behavioral control. Leadership, as a formal system, directly affects the ethical culture of audit firms [69]. Thus, although ethical culture in audit firms tends to be persistent over time making it difficult to control for firm fixed effects, it is likely to change in a significant way around new chief auditor appointment.

Motivated by this pattern, we examine audit quality around the appointments of new chief auditors in audit offices. The subsample consists of audit office-year observations with new chief auditor appointments between 2008 and 2012 to capture a period during which significant changes to ethical culture are likely to occur. For this sample, we include audit office fixed effects in all regressions to account for potential omitted time-invariant firm-specific factors driving results.

The results are reported in column (1) and column (2) of Table 6. We not only control for year and industry fixed effects as in Table 4, but also control for audit office fixed effect. The R-squared tend to be larger than the ones in Table 4, suggesting that audit office fixed effects have additional explanatory power toward explaining audit quality. Most importantly, the coefficients on ethical culture are negative and significant. At the same time, it is interesting to note that some of the control variables such as audit firm size, client importance, and LOCAL are no longer significant, suggesting that audit office fixed effects have absorbed most of their effects on audit quality. In terms of economic effects, a one standard deviation increase in

**Table 5. Robustness test.**

| | (1) | (2) | (3) |
|---|---|---|---|
| | | **Alternative measure** | |
| | | **RE** | |
| | | **\|DA\|** | |
| | *MAO* | *RE* | *\|DA\|* |
| *PHIL* | 3.984** | | |
| | (2.143) | | |
| *phil* | | -0.196** | -0.008*** |
| | | (-2.359) | (-2.925) |
| *ASSET* | -1.044*** | -0.152*** | -0.007*** |
| | (-8.280) | (-3.305) | (-4.610) |
| *LEV* | 0.557 | 0.573 | -0.011 |
| | (0.466) | (1.096) | (-0.673) |
| *ROE* | -0.263 | -0.165 | 0.043* |
| | (-0.404) | (-0.372) | (1.698) |
| *LOSS* | 1.984*** | 0.511*** | 0.040*** |
| | (8.704) | (3.169) | (5.209) |
| *CURRENT* | -0.358*** | -0.015 | -0.002*** |
| | (-2.883) | (-1.412) | (-4.856) |
| *AR* | -0.700 | -0.863 | |
| | (-0.528) | (-1.580) | |
| *INV* | -2.525*** | -0.394 | |
| | (-2.702) | (-1.162) | |
| *BIG4* | 0.901 | -0.949*** | -0.002 |
| | (1.644) | (-3.205) | (-0.297) |
| *LOCAL* | 0.050 | 0.050 | -0.000 |
| | (0.234) | (0.534) | (-0.021) |
| *CLIENT* | 1.078** | 0.186 | -0.013** |
| | (2.156) | (0.803) | (-2.095) |
| *AGE* | 0.079*** | 0.012 | 0.001*** |
| | (3.739) | (1.396) | (2.736) |
| *SOE* | 0.076 | -0.159 | -0.009** |
| | (0.340) | (-1.566) | (-2.552) |
| *SCHOOLTIE* | 0.755 | -0.114 | -0.007 |
| | (1.401) | (-0.363) | (-0.667) |
| *GROWTH* | | | 0.016*** |
| | | | (3.844) |
| *DUAL* | | | -0.001 |
| | | | (-0.166) |
| *INDDIR* | | | 0.021 |
| | | | (0.759) |
| *_cons* | 20.047*** | 1.211 | 0.207*** |
| | (7.360) | (1.152) | (6.221) |
| Year Dummy | | Include | Include |
| Industry Dummy | | Include | Include |
| N | 5,207 | 5,246 | 4,613 |

(*Continued*)

**Table 5.** (Continued)

| | (1) | (2) | (3) |
|---|---|---|---|
| | **Alternative measure** | | |
| | **RE** | | |
| | **\|DA\|** | | |
| | *MAO* | *RE* | *\|DA\|* |
| Pseudo R$^2$/adj. R$^2$ | 33.78% | 2.7% | 27.5% |

Note

*, ** and *** indicate that the regression coefficients are significant at confidence levels of 10%,5% and 1%, respectively. *MAO*, modified audit opinion; *RE*, restate; *DA*, discretionary accruals; *PHIL*, philanthropic contribution of auditor office.

*PHIL* is associated with 5.48% and 0.127% decrease in the incidence of financial restatements, absolute discretionary accruals, respectively.

Overall, the results in column (1) and column (2) of Table 6 show that the coefficients on *PHIL* are negative and significant, indicating that there is a positive and significant relation between ethical culture in audit firms and audit quality even after controlling for audit office fixed effects. Although this finding does not exclude omitted variables that are also changing at the same time as ethical culture and in the same direction, it mitigates concerns that some omitted time-invariant firm-specific variable may be responsible for the documented relations.

We adopt a PSM approach to address the endogeneity arising from the choice of audit office according own characteristics. Specifically, we divided the samples into two groups based on the mean of the ratio of expense of philanthropic contribution on total incomes (0.055), if this ratio beyond 0.055, we defined these samples as philanthropic observations, otherwise are defined as non-philanthropic observations. We estimate the likelihood of future engaging in philanthropic contribution based on the three audit-firm variables in Eq (1) and Eq (2). We then match, without replacement, each philanthropy observation with two non-philanthropy observations with closet engaging in philanthropic contributions probability. We next re-estimate Eq (1) and Eq (2) using the matched sample. The results are shown in column (3) and column (4) of Table 6. We can see that the coefficients for PHIL are -1.825 ($p < 0.05$) and -0.070($p<0.01$), which demonstrates that ethical culture significantly improves auditor independence.

We use the instrumental variable approach to address potential endogeneity concerns in our analyses that there are unobservable variables that affect both audit offices' ethical culture and audit quality. We employ the amount of buddhist temples in the province where the audit office is located as the instrumental variable to perform two-stage regression [70, 71]. Column (5) and column (7) of Table 6 present the results from the first stage of our 2SLS analysis. In line with our expectation, we find a positive and highly significant relation between PHIL and BUD. The results from the second stage of our 2SLS analysis, which reestimates Eq (1) and Eq (2) after replacing our test variables with their predicted values from the first stage regression, are reported in column (6) and column (8) of Table 6. We can see that the coefficients for PHIL(IV) are -12.639 ($p < 0.01$) and -2.078($p<0.01$), which demonstrates that our main inferences are robust to using a 2SLS approach to correct for any endogeneity bias.

## Mechanism of ethical culture in audit firms

As another way to address endogeneity concerns, we test theoretical predictions regarding the inner workings of ethical culture. In the absence of evidence supporting these predictions, it is

**Table 6. Address endogeneity problems.**

| | (1) | (2) | (3) | (4) | (5) | (6) | (7) | (8) |
|---|---|---|---|---|---|---|---|---|
| | A subsample of the appointment of a new chief auditor | | PSM | | Instrumental variable approach | | | |
| | | | | | First stage | Second stage | First stage | Second stage |
| | RE | \|DA\| | RE | \|DA\| | PHIL | RE | PHIL | \|DA\| |
| PHIL | -6.160** | -0.142** | -1.825** | -0.070*** | | | | |
| | (-2.460) | (-2.315) | (-2.082) | (-2.646) | | | | |
| IV:BUD | | | | | 0.002*** | | 0.001** | |
| | | | | | (3.938) | | (2.478) | |
| PHIL(IV) | | | | | | -12.639*** | | -2.078*** |
| | | | | | | (-3.339) | | (-3.515) |
| ASSET | -0.085 | -0.008*** | -0.175*** | -0.009*** | 0.001 | -0.060** | 0.001 | -0.006*** |
| | (-1.412) | (-4.223) | (-3.297) | (-4.399) | (0.849) | (-2.264) | (0.550) | (-4.005) |
| LEV | 0.233 | 0.002 | 0.257 | 0.006 | -0.021** | 0.039 | -0.024** | -0.054** |
| | (0.316) | (0.091) | (0.421) | (0.297) | (-2.229) | (0.142) | (-2.071) | (-2.522) |
| ROE | -0.412 | 0.042 | 0.342 | 0.065** | -0.015* | -0.249 | -0.016* | 0.014 |
| | (-0.678) | (1.566) | (0.695) | (2.211) | (-1.750) | (-1.212) | (-1.814) | (0.528) |
| LOSS | 0.547** | 0.033*** | 0.514*** | 0.039*** | -0.007** | 0.141 | -0.007** | 0.025*** |
| | (2.411) | (3.671) | (2.813) | (4.425) | (-2.169) | (1.269) | (-2.376) | (2.783) |
| CURRENT | 0.002 | -0.001*** | -0.033** | -0.002*** | 0.000 | -0.006 | -0.000 | -0.002*** |
| | (0.145) | (-2.729) | (-2.004) | (-4.153) | (0.305) | (-1.272) | (-0.033) | (-5.057) |
| AR | -0.710 | | -0.681 | | 0.010 | -0.250 | | |
| | (-0.955) | | (-1.103) | | (1.116) | (-0.944) | | |
| INV | 0.245 | | -0.350 | | 0.001 | -0.160 | | |
| | (0.528) | | (-0.903) | | (0.108) | (-0.932) | | |
| BIG4 | -1.176 | 0.012 | -0.534 | 0.016 | -0.029*** | -0.738*** | -0.030*** | -0.048*** |
| | (-0.791) | (0.745) | (-1.111) | (1.153) | (-8.118) | (-6.192) | (-9.828) | (-3.328) |
| LOCAL | 0.135 | 0.007* | -0.119 | -0.001 | -0.003* | -0.005 | -0.003 | -0.005 |
| | (1.008) | (1.871) | (-1.036) | (-0.418) | (-1.776) | (-0.109) | (-1.494) | (-1.567) |
| CLIENT | -1.379 | -0.027 | 0.163 | -0.010 | -0.037*** | -0.338* | -0.035*** | -0.066*** |
| | (-0.690) | (-1.311) | (0.590) | (-1.351) | (-8.556) | (-1.748) | (-6.382) | (-3.745) |
| AGE | 0.000 | 0.000 | 0.016 | 0.001*** | -0.000 | 0.004 | 0.000 | 0.001*** |
| | (0.011) | (1.197) | (1.572) | (2.789) | (-0.125) | (1.094) | (0.142) | (2.743) |
| SOE | -0.030 | 0.003 | -0.151 | -0.012*** | -0.004** | -0.126*** | -0.005* | -0.019*** |
| | (-0.207) | (0.752) | (-1.310) | (-2.978) | (-2.198) | (-2.695) | (-1.840) | (-3.845) |
| SCHOOLTIE | -0.356 | -0.017 | -0.029 | -0.010 | 0.007 | 0.053 | 0.006 | 0.007 |
| | (-0.828) | (-1.243) | (-0.082) | (-0.688) | (1.273) | (0.331) | (1.095) | (0.589) |
| GROWTH | | 0.000 | | -0.000*** | | | 0.000 | 0.001 |
| | | (0.397) | | (-4.135) | | | (0.282) | (0.385) |
| DUAL | | 0.000 | | -0.004 | | | 0.001 | 0.041 |
| | | (0.086) | | (-0.917) | | | (0.406) | (1.449) |
| INDDIR | | -0.000 | | 0.035 | | | 0.011 | 0.007 |
| | | (-0.014) | | (1.035) | | | (0.614) | (0.589) |
| _cons | 1.010 | 0.253*** | 1.733 | 0.233*** | -2.940*** | 1.178** | 0.045* | 0.316*** |
| | (0.536) | (6.109) | (1.424) | (5.479) | (-301.105) | (2.461) | (1.819) | (6.373) |
| Year Dummy | Include | Include | Include | Include | Include | Include | Include | Include |
| Industry Dummy | Include | Include | Include | Include | Include | Include | Include | Include |
| Office Dummy | Include | Include | No | No | No | No | No | No |
| N | 2,904 | 2,719 | 3,479 | 3,414 | 5,246 | | 4,613 | |

(*Continued*)

**Table 6.** (*Continued*)

| | (1) | (2) | (3) | (4) | (5) | (6) | (7) | (8) |
|---|---|---|---|---|---|---|---|---|
| | A subsample of the appointment of a new chief auditor | | PSM | | Instrumental variable approach | | | |
| | | | | | First stage | Second stage | First stage | Second stage |
| | *RE* | *\|DA\|* | *RE* | *\|DA\|* | *PHIL* | *RE* | *PHIL* | *\|DA\|* |
| Pseudo R² /adj. R² | 6.8% | 27.7% | 2.8% | 25.3% | | | 9.4% | 27.8% |

Note

*, ** and *** indicate that the regression coefficients are significant at confidence levels of 10%,5% and 1%, respectively.

*RE*, restate; *DA*, discretionary accruals; *PHIL*, philanthropic contribution of auditor office.

likely that audit firms' ethical culture is not responsible for the results. Theoretically, audit firms can operate through two channels through their culture. Firstly, ethical culture in audit firms can be used as a selection mechanism to select and attract auditors with similar values to the firm, who act based on their own values. Secondly, ethical culture in audit firms can affect auditor behavior directly through group norms.

## Ethical culture in audit firms as a selection mechanism

An important characteristic of corporate culture is that it is shaped by the attraction-selection-attrition cycle [30], where "attraction to an organization, selection by it, and attrition from it yield particular kinds of persons in an organization. These people determine organizational behavior." In line with this concept of corporate culture as a selection mechanism, empirical evidence suggests that individuals like to join firms with cultures similar to their own and are less likely to be satisfied if their values are incompatible with those of the firm [72–74].

In Table 7, we test whether the ethical culture in audit firms acts as a selection mechanism in ways that are consistent with predictions from the theoretical literature. In column (1) to column (8), we study the attraction and selection part of the ASA process in 2010 and 2011, respectively. In addition, we divide the sample into audit office with a high ethical culture and audit office with a low ethical culture based on whether the office's philanthropic contributions is above the sample mean. If the office's philanthropic contributions is higher than the sample mean, we define it as an audit office with high ethical culture. If the office's philanthropic contributions is lower than the sample mean, we define it as an audit office with low ethical culture. If ethical culture acts as a selection mechanism, then the prediction is that the audit quality of auditors who chose job-hopping from audit offices with low (high) ethical culture to audit offices with high (low) ethical culture is significantly higher (lower) than audit quality of the rest of auditors.

The test is conducted at the auditor level, where *HOPPING1* equal to one, if auditors chose job-hopping from audit offices with low ethical culture to audit offices with high ethical culture in given year, and 0 otherwise. In addition, we control for *HOPPING2*, which equal to one if auditors chose job-hopping from audit offices with low ethical culture to audit offices with low ethical culture in given year, and 0 otherwise. And we also control for industry fixed effects. In column (1) to column (4), the coefficient for *HOPPING1* are -0.106 ($p<0.01$), -0.028($p<0.05$), -0.088($p<0.05$) and -0.017 ($p<0.1$), respectively, which are negative and statistically significant. These findings are consistent with the theoretical prediction that audit offices with a high ethical culture attract auditors that share similar values and beliefs and these auditors have significantly higher audit quality than other auditors.

**Table 7. Audit firm's culture as a selection mechanism.**

| | RE | \|DA\| | RE | \|DA\| | RE | \|DA\| | RE | \|DA\| |
|---|---|---|---|---|---|---|---|---|
| | (1) | (2) | (3) | (4) | (5) | (6) | (7) | (8) |
| HOPPING1 | -0.106** | -0.028** | -0.088** | -0.017* | | | | |
| | (-2.405) | (-1.982) | (-2.191) | (-1.735) | | | | |
| HOPPING3 | | | | | 0.242** | 0.061** | 0.512** | 0.155* |
| | | | | | (2.225) | (2.217) | (2.311) | (1.899) |
| ASSET | -0.011 | -0.007* | -0.021** | 0.002 | -0.010 | -0.014** | -0.010 | -0.002 |
| | (-0.819) | (-1.710) | (-2.234) | (0.652) | (-0.661) | (-2.445) | (-0.697) | (-0.424) |
| LEV | 0.039 | -0.071* | 0.077 | -0.052* | -0.011 | 0.071 | 0.093 | 0.214*** |
| | (0.252) | (-1.784) | (0.645) | (-1.676) | (-0.070) | (1.114) | (0.430) | (2.989) |
| ROE | -0.179 | 0.076 | 0.016 | 0.078** | -0.103 | -0.300*** | 0.034 | 0.042 |
| | (-0.924) | (1.606) | (0.138) | (2.015) | (-0.514) | (-2.879) | (0.178) | (0.796) |
| LOSS | 0.098 | 0.046** | 0.048 | 0.055*** | 0.000 | -0.015 | 0.142 | 0.006 |
| | (1.043) | (2.224) | (0.924) | (3.341) | (0.001) | (-0.392) | (1.473) | (0.306) |
| CURRENT | 0.003 | 0.001 | -0.002 | -0.002*** | -0.004 | -0.002 | 0.002 | -0.001 |
| | (0.880) | (0.391) | (-0.983) | (-3.676) | (-0.975) | (-1.001) | (0.496) | (-1.594) |
| AR | -0.051 | | -0.239* | | 0.316 | | -0.011 | |
| | (-0.261) | | (-1.901) | | (1.371) | | (-0.058) | |
| INV | -0.013 | | -0.075 | | -0.120 | | -0.015 | |
| | (-0.113) | | (-0.901) | | (-1.198) | | (-0.149) | |
| LOCAL | 0.004 | -0.001 | 0.039* | 0.012* | 0.052 | 0.007 | 0.060* | 0.001 |
| | (0.143) | (-0.127) | (1.698) | (1.901) | (1.341) | (0.589) | (1.824) | (0.113) |
| CLIENT | 0.037 | -0.030 | 0.027 | -0.015 | 0.155 | -0.081*** | -0.080 | -0.044** |
| | (0.331) | (-1.284) | (0.547) | (-1.276) | (1.439) | (-3.176) | (-1.339) | (-2.462) |
| AGE | 0.002 | -0.001 | 0.002 | 0.001* | 0.002 | 0.002 | -0.002 | 0.001 |
| | (0.633) | (-0.806) | (0.781) | (1.886) | (0.615) | (1.498) | (-0.707) | (1.389) |
| SOE | -0.030 | -0.022** | -0.054** | -0.007 | -0.013 | -0.029** | -0.004 | -0.029** |
| | (-0.867) | (-2.036) | (-2.205) | (-1.038) | (-0.355) | (-2.068) | (-0.112) | (-2.177) |
| SCHOOLTIE | 0.009 | 0.093* | -0.006 | -0.028 | 0.067 | -0.046*** | 0.033 | 0.020 |
| | (0.069) | (1.746) | (-0.100) | (-1.282) | (0.497) | (-2.681) | (0.317) | (0.300) |
| GROWTH | | 0.011 | | 0.003 | | 0.008 | | 0.001*** |
| | | (1.236) | | (0.382) | | (1.646) | | (12.172) |
| DUAL | | -0.005 | | -0.012 | | 0.020 | | -0.018 |
| | | (-0.355) | | (-0.231) | | (1.358) | | (-1.618) |
| INDDIR | | 0.057 | | -0.028 | | 0.070 | | 0.001 |
| | | (0.932) | | (-1.282) | | (0.691) | | (0.015) |
| HOPPING2 | -0.168*** | -0.009 | 0.041 | -0.017 | | | | |
| | (-5.315) | (-0.318) | (0.761) | (-1.245) | | | | |
| HOPPING4 | | | | | 0.029 | -0.030 | -0.179*** | -0.066*** |
| | | | | | (0.224) | (-0.945) | (-3.820) | (-3.711) |
| _cons | 0.205 | 0.244*** | 0.580*** | 0.049 | 0.391 | 0.393*** | 0.358 | 0.149 |
| | (0.745) | (3.054) | (2.689) | (0.906) | (1.107) | (3.102) | (1.050) | (1.177) |
| Year = 2010 | Yes | Yes | Yes | Yes | | | | |
| Year = 2011 | | | | | Yes | Yes | Yes | Yes |
| Industry Dummy | Include | Include | Include | Include | Include | Include | Include | Include |
| N | 652 | 560 | 1,381 | 1,224 | 455 | 381 | 592 | 493 |

(*Continued*)

**Table 7.** (Continued)

| | *RE* | *\|DA\|* | *RE* | *\|DA\|* | *RE* | *\|DA\|* | *RE* | *\|DA\|* |
|---|---|---|---|---|---|---|---|---|
| | (1) | (2) | (3) | (4) | (5) | (6) | (7) | (8) |
| Pseudo R²/adj. R² | 4.7% | 13.2% | 3.1% | 34.1% | 9% | 19.7% | 6% | 13.2% |

Note

*, ** and *** indicate that the regression coefficients are significant at confidence levels of 10%,5% and 1%, respectively. *RE*, restate; *DA*, discretionary accruals; *HOPPING1*, where HOPPING1 equal to one if auditors chose job-hopping from audit offices with low ethical culture to audit offices with high ethical culture, and 0 otherwise; *HOPPING2*, which equals one if any one of signing auditors who chose job-hopping from audit offices with high ethical culture to audit offices with low ethical culture in a given year, and 0 otherwise.

In column (5) to column (8), we examine the attrition part of ASA process that auditors who are not compatible with firm's culture were discharged from the audit offices in 2010 and 2011, respectively. To test this prediction, the explanatory variable is *HOPPING3*, which equals one if any one of signing auditors who chose job-hopping from audit offices with high ethical culture to audit offices with low ethical culture in a given year, and 0 otherwise. Besides, we control for variable *HOPPING4*, which equal to one if auditor chose job-hopping audit offices with high ethical culture to audit offices with high ethical culture in a given year, and 0 otherwise. We also control for industry fixed effects. In column (5) to column (8), the coefficient on explanatory variable *HOPPING3* are 0.242 ($p < 0.05$), 0.061($p < 0.05$), 0.512($p < 0.05$) and 0.155 ($p < 0.1$), which are positive and significant, consistent with the attrition prediction that the audit quality of auditors who were discharged by audit offices is significantly lower than the audit quality of the rest of auditors, and audit offices discharge these auditors who are not compatible with their firm culture.

### Ethical culture in audit firms acting through group norms

The previous section presents evidence that is consistent with corporate culture acting as a selection mechanism. As a result, corporate culture attracts or selects auditors who share the same values with the organization, and they act according to their own internal norms. As opposed to purely external sanctions, such as material rewards or punishments, an internal norm refers to a pattern of behavior shaped by one's value system [75]. Thus, signing auditors with high ethical beliefs are inclined to maintain objectivity and independence, and thus improve the quality of their audits.

The theoretical literature [37] suggests that corporate culture influences both individual behavior and group norms through internal norms, which act as a selection mechanism. Unlike internal norms, group norms are enforced by members through rewards and punishments. Corporate culture is the prevailing group norm in corporations.

In Table 8, using data from a subsample of clients audited by job-hopping auditors, we examine whether group norms influence auditors' behavior. For this sample, we not only control for time-fixed effects and industry-fixed effects, but we control for auditor individual fixed effects, removing the effect of internal norms when they do not change over time. In column (1) and column (2) of Table 8, the key coefficients are 1.867($p < 0.01$) and -0.428($p < 0.01$), respectively, which indicate that an auditor working in an audit office with high ethical culture is more likely to constrain client's financial restatement and earnings management than the same working in a firm with low ethical culture, suggesting that ethical culture have significant impact on auditor's individual behavior.

**Table 8. The direct effect of audit office's culture on auditor's behavior.**

|  | *RE* | *\|DA\|* |
|---|---|---|
|  | **(1)** | **(2)** |
| *PHIL* | -1.867*** | -0.428*** |
|  | (-3.304) | (-3.230) |
| *ASSET* | -0.021 | -0.004 |
|  | (-0.933) | (-0.434) |
| *LEV* | -0.122 | -0.083 |
|  | (-0.403) | (-0.945) |
| *ROE* | 0.204 | 0.017 |
|  | (0.759) | (0.235) |
| *LOSS* | 0.281*** | 0.003 |
|  | (2.644) | (0.115) |
| *CURRENT* | -0.003 | -0.001 |
|  | (-0.964) | (-1.279) |
| *AR* | -0.168 |  |
|  | (-0.730) |  |
| *INV* | 0.008 |  |
|  | (0.038) |  |
| *BIG4* | -0.007 | 0.088 |
|  | (-0.020) | (0.968) |
| *LOCAL* | 0.004 | 0.003 |
|  | (0.072) | (0.201) |
| *CLIENT* | 0.091 | 0.005 |
|  | (0.262) | (0.106) |
| *AGE* | -0.001 | -0.001 |
|  | (-0.148) | (-0.606) |
| *SOE* | -0.039 | 0.015 |
|  | (-0.573) | (0.838) |
| *SCHOOLTIE* | -0.129 | -0.047* |
|  | (-1.346) | (-1.696) |
| *GROWTH* |  | 0.001 |
|  |  | (0.669) |
| *DUAL* |  | -0.005 |
|  |  | (-0.380) |
| *INDDIR* |  | 0.082 |
|  |  | (0.618) |
| *_cons* | 0.635 | 0.087 |
|  | (1.035) | (0.455) |
| Year Dummy | Include | Include |
| Industry Dummy | Include | Include |
| Auditor Dummy | Include | Include |
| N | 629 | 554 |
| Pseudo $R^2$/adj. $R^2$ | 41.9% | 60.1% |

Note

*, ** and *** indicate that the regression coefficients are significant at confidence levels of 10%,5% and 1%, respectively. *RE*, restate; *DA*, discretionary accruals; *PHIL*, philanthropic contribution of auditor office.

## Conclusion

While the impact of auditor characteristics on audit quality have been studied extensively, relatively little is known about the role of corporate culture of audit firms in influencing audit quality at the audit office level. In this paper, we use the size-adjusted anonymous philanthropic contributions of each auditor office in China and examine how ethical culture in audit offices influence their audit quality.

We measure ethical culture of audit firm as the ratio of expense of the philanthropic contribution on the total income of each audit office. Using 5246 audits in the Chinese market between 2010 and 2012, we find that the level of ethical culture in audit firms is significantly negatively associated with the magnitude of earnings management and the frequencies of financial restatements of their client firms. The effects are also economically significant: one standard deviation (0.905) increase in the audit firm's culture is associated with a decrease in earnings management of 0.076% and a decrease in the presence of accounting errors of 0.849%. We also find this association is even stronger when auditors provide services to clients that are economically important or bear a social connection with them.

Further evidence show that the ethical culture can both act as a mechanism that attract auditors with a compatible internal norm, and great a group norm in audit firms that directly shape auditor behavior. Collectively, my study suggests that ethical culture of audit firms can significantly improve audit quality.

Overall, a firm's culture is an important determinant of its audit quality, according to this study. In an effort to improve auditor independence, regulators are increasingly focusing on audit firms' cultures. In an effort to improve auditor independence, regulators are increasingly focusing on audit firms' cultures. Audit culture and its evolution can help us better understand an audit firm's internal dynamics and impact on audit quality.

## Author Contributions

**Conceptualization:** Yiling Zhang.

**Formal analysis:** Yiling Zhang.

**Methodology:** Lang Wei.

**Writing – original draft:** Yiling Zhang.

**Writing – review & editing:** Lang Wei.

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
