## [Decision Letter · Decision Letter 0]

15 Jun 2022

PONE-D-22-12398Philanthropy, Audit Firms Culture and Auditor IndependencePLOS ONE

Dear Dr. Wei,

Thank you for submitting your manuscript to PLOS ONE. After careful consideration, we feel that it has merit but does not fully meet PLOS ONE’s publication criteria as it currently stands. Therefore, we invite you to submit a revised version of the manuscript that addresses the points raised during the review process.

We look forward to receiving your revised manuscript.

Kind regards,

Pattanaporn Chatjuthamard, Ph.D.

Academic Editor

PLOS ONE

Journal Requirements:

Reviewers' comments:

Reviewer's Responses to Questions

**Comments to the Author**

1. Is the manuscript technically sound, and do the data support the conclusions?

Reviewer #1: Yes

Reviewer #2: Yes

Reviewer #3: Partly

2. Has the statistical analysis been performed appropriately and rigorously? 

Reviewer #1: Yes

Reviewer #2: Yes

Reviewer #3: Yes

3. Have the authors made all data underlying the findings in their manuscript fully available?

Reviewer #1: Yes

Reviewer #2: Yes

Reviewer #3: Yes

4. Is the manuscript presented in an intelligible fashion and written in standard English?

Reviewer #1: No

Reviewer #2: Yes

Reviewer #3: Yes

5. Review Comments to the Author

Reviewer #1: I liked the research question of the manuscript. It has a clear research question, adequate methodology, and novel results. What is needed at this point is carefully relating results to the already existing papers and make the paper more consistent throughout the paper. The manuscript can be substantially improved in this direction.

First, there is no connection between the literature review section and the results section. More discussion of the result, as well as literature support for the result, is required. For example, whether or not this outcome is consistent with other literature.

Second, there are several typo in the manuscript e.g. page 15 “companies in 2003,, whereas only 25% of Chinese listed firms hired Top-10 audit firms” or “such as social ties to conduct business,” or page 32 “deviation increase in a firm’s ethical culture is associated with an increase in the audit quality, by 0.844%,14.77% (17.26%).” Besides, there are some incorrectly citation e.g. (Shimin Chen et al., 2010). Furthermore, the authors should watch out for proper space between words. Lastly, the author may consider to use comma when the number is more than thousand.

Third, though authors ran several tests and robustness, there is still lack of consistent variable that authors used. In 5.3.2 Potential endogeneity in the match between audit firms and client’s section, the author used the mean of the ratio of expense of philanthropic contribution on total incomes to divided the sample instead of the median. Whereas, in previous test the authors usually use median instead. Would we get the same result if we use median instead? Similarly, to test hypothesis H2, what happen if you divided client based on mean instead of median.

Reviewer #2: 1. The abstract should include the finding on H3 even thought it was not statistically significant but the fact should be reported.

2. Table 7 result is important and it should be part of the main results of this study as it uses the direct proxy of audit quality which I believe it is a better proxy as compared to other indirect proxies of audit quality such as RE or |DA|.

3. The results on H2 (See Table 5) are not strong as they are significant at 10%. Thus, they should be mentioned on the inference section regarding the 10% significance.

Reviewer #3: • Table 4 to 6 can be combined into a single and more meaningful table

• Table 7 changes the format of reporting the result. Please uses a consistent format such as in Table 4 to 6

• Table 8 mentions about alternative measure of ethical culture. However, this is not clear in the table. Please provide more explanation to make each table self-explanatory.

• Table 9 mentions about addressing omitting variables. However, this is not clear in the table. Please provide more explanation to make each table self-explanatory.

• Table 10 mentions about addressing potential endogeneity issues. However, this is not clear in the table. Please provide more explanation to make each table self-explanatory.

• There are several typos (e.g., “great a group norm” or “reasonable homogeny”) and grammar mistakes. Please correct them.

• Not clear how the paper classify high ethical culture and low ethical culture.

• What is the meaning of absolute value in the variable name “DA” (i.e., || ). Why can’t the variable name be consistent with other variables.

• Please explain the modified Jones model (Dechow and Sloan, 1995) to estimate discretionary accruals

• The following part is not clear “However, since Kothari, Leone, and Wasley (2005) find that performance-matched discretionary accrual measures enhance the reliability of the inferences from earnings management research, we control for the company’s prior performance and compute discretionary accrual (||).”

• Please improve the economic interpretation of the estimated results (e.g., indicating that a one standard deviation (0.905) increase in the audit firm’s culture is associated with a decrease in the presence of accounting errors of 0.849%, which is 5.4% of the mean financial restatements rate of 15.6%) as the current explanation is not clear and hard to understand. Also, in the conclusion, “0.844%,14.77% (17.26%)”, this is not clear at all.

• The use of alternative measures of ethical culture that is simply a conversion of originally continue variables into a dummy variable is trivial.

• In this part, “We also find support for this prediction in this section. Thus, although ethical culture in audit firms tends to be persistent over time making it difficult to control for firm fixed effects, it is likely to change in a significant way around new chief auditors appointments.”, it would be more evident if you can show that firms with new chief auditors have different ethical culture as measured by your key variable. Furthermore, using a sub-sample analysis does not fully address the endogeneity issues. It would be important to consider instrumental variable analysis.

• “Overall, the results in table 9 show that there is a positive and significant relation” Why positive?

• Please use a consistent variable name in the job-hopping part. Is HOPPING the same as hopping? Also HOPPING, HOPPING_1, hopping, and HOPPING_2 are not explained in Table 2. Thus, the discussion of the result in this part is still not clear.

• The sub-sample analysis in Table 12 is not clearly explained.

6. PLOS authors have the option to publish the peer review history of their article (what does this mean?). If published, this will include your full peer review and any attached files.

Reviewer #1: No

Reviewer #2: No

Reviewer #3: No

---

## [Author Response · Author response to Decision Letter 0]

20 Jul 2022

Dear editor,

Our most sincere thanks to you, Reviewer 1, Reviewer 2, and Reviewer 3 for the very thoughtful review and helpful comments, and suggestions on our writing in this round. 

We have revised the literature review and hypothesis sections of the paper to highlight the theoretical logic of the paper. We have also revised the writing details and formatting of the article based on the reviewers' suggestions, which improved the quality of the article. With your thoughtful comments, our latest revision has improved a lot.

Below, we provide the original comments from Reviewer 1, Reviewer 2 and Reviewer 3 in boldface and our responses to each item in regular font. We do this point by point.

We highlight the main amendment in blue in the revised manuscript.

Response to Reviewer 1

 ( PLOS-D-22-12398)

First, there is no connection between the literature review section and the results section. More discussion of the result, as well as literature support for the result, is required. For example, whether or not this outcome is consistent with other literature.

Thanks for your helpful suggestion.

Inspired by this comment, we rewrite the literature review and hypothesis development section. In this revised manuscript, first, we define culture from existing kinds of literature and summarize relevant research on the impact of culture on auditors’ professional judgment. Second, we propose reasons why there is a lack of research on the impact of firm culture on audit quality. At the same time, we summarize research in the corporate governance literature that uses corporate giving to measure ethical culture or pro-social culture in corporate, and the rationale for doing so. Third, we propose the hypothesis of this paper based on the relevant literature.

Then, in this section, we remove the literature that was not relevant to the result.

We rewrite the literature review and hypothesis development as follow:

Literature review and hypothesis development 

Culture, broadly defined, refers to “traditions, customs, moral values, religious beliefs, and all other forms of behavior that have passed the test of time” (Pejovich 1999,166). Human behavior is normalized through these unwritten rules because they provide society members with schematic, mental models of what is “good” versus “bad”, what is “appropriated” versus “inappropriate”, and they set norms for behavior (Parboteeah et al., 2015). According to this idea, auditors, as members of society, are not only more likely to be aware of social rules, but they are also more likely to expect that others will follow those rules as well. In other words, auditors’ behavior, especially for their professional judgments, are likely to be affected by the schematic, mental model prevailing in a society (Patel et al., 2002; Nolder and Kadous, 2017). Prior research finds cultural differences in auditors' compliance with company policies affect their assessment procedures for fraud risk (Bik and Hooghiemstra, 2017). Specifically, they find that collectivism and societal trust are negatively associated with compliance with global firm policy, while religiosity is positively associated with compliance. Nolder and Riley (2014) review the audit literature that the association culture with variation in auditor’s ethical judgment. And it is proposed that researchers should pay attention to the effect of audit firm culture on auditor’s judgment and decision-making.

There is growing literature that examines the effect of auditor’s characteristics on audit quality (e.g., Lennox and Pittman, 2010). However, there has been little study of the role of audit firm culture and ethics on audit quality. The reason for this is partly due to the difficulty of developing an empirical measure of audit firm culture and ethics.

 Research on corporate governance addresses this difficulty by measuring firm culture based on corporate philanthropy (Bereskin et al., 2013). According to Schein (1985), “artifacts” are visible manifestations of an organization’s underlying values. To the extent that philanthropy can be seen as an artifact of a firm's culture, it can reflect socially responsible values and influence the propensity for wrongdoing. 

Corporate giving has been extensively studied in a wide variety of disciplinary contexts. The primary reason for corporate giving is that it can enhance a firm's public image and generate a positive image of the company. Fisman et al. (2006) models corporate philanthropy as a signal of trust where product quality is intangible. In addition, a firm's corporate giving is likely to be associated with a culture of high regard for its reputation if it is motivated by a desire for high integrity and building trust with customers. Therefore, these firms doing giving are reluctant to engage in misconduct since they spend a lot of time and money building their image, and engaging in misconduct comes at a higher price (Bea and Cameron, 2006; Varadarajan and Menon, 1988). Therefore, audit quality is the output product for the audit firms, when stakeholders, such as regulators, are unable to judge audit quality, the audit firm's giving can serve as a signal of a high-audit quality. Moreover, since donation audit firms have spent a great deal of time and energy building reputations, they will be more adamant about maintaining professional independence to prevent reputation losses caused by audit failures.

In addition to the above literature, there are also views that corporate giving as a manifestation of corporate social responsibility stems from an altruistic or pro-social behavior (Campbell et al., 1999). Brown and Ferris (2004) stated that “selfless or not, these acts involve a degree of compassion and commitment to others.” The auditor's prosocial ethics may influence his behavior and reduce his willingness to tolerate clients' opportunistic behavior.

Such a role for personal ethics in the firm’s propensity to engage in wrongdoing is also supported by recent papers. Biggerstaff et al. (2012) find that Executives who are unethical are more likely to manipulate earnings. Davidson et al. (2012) track unethical behavior based on past legal infractions, like driving under the influence. Using this measure, they find that executive misconduct is more likely to occur. If corporate giving is a manifestation of the auditor’s pro-social ethic, it is likely to increase their propensity to insist on professional independence. This leads us to our first hypothesis,

H1: Ethical culture in audit firms, measured as their philanthropic contributions over total incomes, is positively associated with audit quality.

Second, there are several typo in the manuscript e.g. page 15 “companies in 2003,, whereas only 25% of Chinese listed firms hired Top-10 audit firms” or “such as social ties to conduct business,” or page 32 “deviation increase in a firm’s ethical culture is associated with an increase in the audit quality, by 0.844%,14.77% (17.26%).” Besides, there are some incorrectly citation e.g. (Shimin Chen et al., 2010). Furthermore, the authors should watch out for proper space between words. Lastly, the author may consider to use comma when the number is more than thousand.

Thanks for your suggestion!

I correct the details of the error one by one. 

(1) While considering the previous comment, we remove “companies in 2003, whereas only 25% of Chinese listed firms hired Top-10 audit firms” or “such as social ties to conduct business”.

(2) We check the empirical results, and rewrite “deviation increase in a firm’s ethical culture is associated with an increase in the audit quality, by 0.844%,14.77% (17.26%)” as follows,

The effects are also economically significant: one standard deviation (0.905) increase in the audit firm’s culture is associated with a decrease in earnings management of 0.076% and a decrease in the presence of accounting errors of 0.849%.

(3) We seriously corrected my erroneous citation as follow,

Chen et al., 2010

(4) We have used commas for numbers over thousands in articles.

Third, though authors ran several tests and robustness, there is still lack of consistent variable that authors used. In 5.3.2 Potential endogeneity in the match between audit firms and client’s section, the author used the mean of the ratio of expense of philanthropic contribution on total incomes to divided the sample instead of the median. Whereas, in previous test the authors usually use median instead. Would we get the same result if we use median instead? Similarly, to test hypothesis H2, what happen if you divided client based on mean instead of median.

Thanks for your helpful suggestion.

In order to keep the consistency of the variables, we redefine the variable Client importance, the indicator variable that equals 1 if the importance of client i in yeat t is greater than the sample mean for auditor office j, and 0 otherwise. And we again regress the model (3) and model (4), and the empirical results are as follows,

Then we replace table5 with this result.

In addition, in 5.3.2 Potential endogeneity in the match between audit firms and client’s section, we use the median instead of the mean to test again, the results are shown in table 1, which is still robust.

Response to Reviewer 2

( PLOS-D-22-12398)

1. The abstract should include the finding on H3 even though it was not statistically significant but the fact should be reported.

Thanks for your helpful suggestion.

In the abstract section, we did not articulate the empirical conclusion about H3. Therefore, we check the empirical result on H3, which shows that the impact of ethical culture on improving audit quality is pronounced for clients where at least one senior executive of the client firm has school ties to any one of the contracted auditors. And we rewrite the finding on H3 in the abstract as follow,

We also find this association is even stronger when auditors provide services to clients that are economically important or when signing auditors bear school ties with at least one top executive of the client.

2. Table 7 result is important and it should be part of the main results of this study as it uses the direct proxy of audit quality which I believe it is a better proxy as compared to other indirect proxies of audit quality such as RE or |DA|.

Thanks for your helpful suggestion. 

I agree with your point of view. Audit opinions are very direct measures of audit quality because the audit opinion is the auditor’s responsibility and directly under his or her influence and control. In China, a higher propensity to issue MAOs could suggest the auditor possesses a higher audit quality. More importantly, the MAOs formulation process is a setting that allows direct insights into auditor independence. Therefore, we considered Maos as the main explained variable in this study during the study design.

However, Maos, as the main explained variable in this study, have several limitations on the results of the study. One is that, MAOs are relatively rare. This reduces statistical power in tests using large samples of healthy firms. According to our descriptive statistics for MAOs in our article sample, the percentage of firm-year observations that received MAOs is 3.94% over the sample period, which is consistent with Ku et al. (2016). Because the percentage of Maos is too rare, when we empirically test the moderating effect of school ties, we cannot obtain statistical results that are in line with expectations. Second is that, while MAOs are the most direct measurement of audit quality, more MAOs might instead that auditors are over-conservative, which arguably would reduce audit quality (Defond and Zhang, 2014). To overcome the above shortcomings, we employ restatement and earning management to proxy for audit quality. At the same time, we also used MAOs as the explained variable to re-test the main regression, and the results were robust.

3. The results on H2 (See Table 5) are not strong as they are significant at 10%. Thus, they should be mentioned on the inference section regarding the 10% significance.

Thank you for your suggestion.

We take your comments into consideration and incorporated those by Reviewer1, we redefine the variable Client importance, the indicator variable that equals 1 if the importance of client i in yeat t is greater than the sample mean for auditor office j, and 0 otherwise. And we again regress the model (3) and model (4), and the empirical results are as follows, the coefficients for PHIL is -2.012(-0.058) and is significant at p<0.05 (p<0.005), and the coefficients for PHIL*CLIENT is -6.154(-0.114) and is significant at p<0.01(p<0.05). Therefore, we replace table5 with this result.

Response to Reviewer 3

( PLOS-D-22-12398)

1.Table 4 to 6 can be combined into a single and more meaningful table.

Thanks for your helpful suggestion!

We reassembled Tables 4 to 6 into one large table named Table 4 as follow, it looks better!

2.Table 7 changes the format of reporting the result. Please uses a consistent format such as in Table 4 to 6.

Thanks for your comments!

I have adjusted the format of Table 7 to be consistent with the previous table format, and we have combined it with Table 8 into a new table named Table 5. It looks better now!

3.Table 8 mentions about alternative measure of ethical culture. However, this is not clear in the table. Please provide more explanation to make each table self-explanatory.

Thanks for your suggestion!

We have revised the format of Table 8 and combined with the previous Table 7 into a new Table 5 based on your previous comment to make the table clearer. In addition, we have added an explanation of the surrogate variables in the article, as follows,

Robustness tests were performed using mean and median as surrogate variables for ethical culture and defined as phil. The results are shown in columns (2) to column (5) of Table 5.

4.Table 9 mentions about addressing omitting variables. However, this is not clear in the table. Please provide more explanation to make each table self-explanatory.

Thanks for your suggestion!

We have revised the format of Table 9 and combined it with the previous Table 10 into a new Table 6 to make the table clearer.

5.Table 10 mentions about addressing potential endogeneity issues. However, this is not clear in the table. Please provide more explanation to make each table self-explanatory.

Thanks for your suggestion!

Combined with the previous comment, I have revised Table 10, and combined it with the previous Table 9 into a new Table 6, it looks better now!

6.There are several typos (e.g., “great a group norm” or “reasonable homogeny”) and grammar mistakes. Please correct them.

Thank for your detailed comment! 

We have corrected it.

7.Not clear how the paper classify high ethical culture and low ethical culture.

Thanks for your suggestion!

We redefine the distinction between high and low moral cultures, as follows,

In addition, we divide the sample into audit office with a high ethical culture and audit office with a low ethical culture based on whether the audit offce’s philanthropic contributions is above the sample mean. If the office’s philanthropic contributions is higher than the sample mean, we define it as an audit offce with high ethical culture. If the offce’s philanthropic contributions is lower than the sample mean, we define it as an audit offce with low ethical culture.

8.What is the meaning of absolute value in the variable name “DA” (i.e., || ). Why can’t the variable name be consistent with other variables.

Thanks for your suggestion!

We have modified DA to be consistent with other variables, as follows,

In equation (2), the dependent variable DA, discretionary accrual, we employ performance-adjusted discretionary accruals as proxy variables for audit quality using the model suggested by Kothari et al. (2005).

9. Please explain the modified Jones model (Dechow and Sloan, 1995) to estimate discretionary accruals

Thanks for your helpful suggestion!

In the research design section, we add a paragraph that explains how to estimate discretionary accruals using the modified Jones model(Dechow and Sloan, 1995), as follows,

Second, in equation (2), the dependent variable DA, discretionary accrual, we employ performance-adjusted discretionary accruals as proxy variables for audit quality using the model suggested by Kothari et al. (2005), which is computed as follow. For each two-digit SIC code industry and year with a minimum of 10 observations, we estimate the cross-sectional version of the modified Jones model in Eq (3). Residuals from Eq (3) are DA before adjusting for firm performance.

10. The following part is not clear “However, since Kothari, Leone, and Wasley (2005) find that performance-matched discretionary accrual measures enhance the reliability of the inferences from earnings management research, we control for the company’s prior performance and compute discretionary accrual (||).”

Thanks for your helpful suggestion!

In combination with the previous comment, we have rewritten this part.

11.Please improve the economic interpretation of the estimated results (e.g., indicating that a one standard deviation (0.905) increase in the audit firm’s culture is associated with a decrease in the presence of accounting errors of 0.849%, which is 5.4% of the mean financial restatements rate of 15.6%) as the current explanation is not clear and hard to understand. Also, in the conclusion, “0.844%,14.77% (17.26%)”, this is not clear at all.

Thanks for your helpful suggestion!

We have rewritten the sentence as follows “a one standard deviation (0.905) increase in the audit firm’s culture is associated with a decrease in the presence of accounting errors of 0.849%, which is 5.4% of the mean financial restatements rate of 15.6%)”.

In terms of economic effect, an increase in the audit office's ethical culture of one standard deviation (0.905) is related with a drop in the presence of accounting error of 1.849%, which is 11.9 % of the mean financial restatements rate of 15.6 %.

In the conclusion section, we have written the sentence “0.844%,14.77% (17.26%)”, as follows,

The effects are also economically significant: one standard deviation (0.905) increase in the audit firm’s culture is associated with a decrease in earnings management of 0.076% and a decrease in the presence of accounting errors of 1.849%.

12.The use of alternative measures of ethical culture that is simply a conversion of originally continue variables into a dummy variable is trivial.

Thanks for your helpful suggestion!

In the robustness section, we considered your suggestion and re-selected proxy variables for surrogate ethical culture for robustness testing, as follows,

We calculated the mean and median foundation giving per audit office for the entire sample period from 2010 to 2012. Robustness tests were performed using mean and median as surrogate variables for ethical culture and defined as phil. The results showing in columns (2) to column (5) of Table 5 again confirm the robustness of our results.

13. In this part, “We also find support for this prediction in this section. Thus, although ethical culture in audit firms tends to be persistent over time making it difficult to control for firm fixed effects, it is likely to change in a significant way around new chief auditors appointments.”, it would be more evident if you can show that firms with new chief auditors have different ethical culture as measured by your key variable. Furthermore, using a sub-sample analysis does not fully address the endogeneity issues. It would be important to consider instrumental variable analysis.

Thanks for your helpful suggestion!

Firstly, we provide additional theoretical evidence from the literature that audit offices with new chief auditors have different ethical cultures, as follows,

The chief aditor of the audit office is the main leader in charge of the office and is responsible for planning and managing the firm's various audit engagements and teams. Trevino et al. (1998) find that ethical culture can be regarded as a subset of organizational culture that represents the multidimensional interplay among formal and informal systems of behavioral control. Leadership, as a formal system, directly affects the ethical culture of audit firms. Thus, although ethical culture in audit firms tends to be persistent over time making it difficult to control for firm fixed effects, it is likely to change in a significant way around new chief auditor appointment.

Secondly, we consider your helpful suggestion and use the instrumental variable approach to address potential endogeneity concerns in our analyses that there are unobservable variables that affect both audit offices’ ethical culture and audit quality. We employ the amount of buddhist temples in the province where the audit office is located as the instrumental variable to perform two-stage regression. The results are shown in column (5) to column (8) of Table6, which demonstrates that our main inferences are robust to using a 2SLS approach to correct for any endogeneity bias.

14.“Overall, the results in table 9 show that there is a positive and significant relation” Why positive?

Thanks for your helpful comments!

We correct this sentence “Overall, the results in table 9 show that there is a positive and significant relation” as follows,

Overall, the results in column (1) and column (2) of Table 6 show that the coefficients on PHIL are negative and significant, indicating that there is a positive and significant relation between ethical culture in audit firms and audit quality even after controlling for audit office fixed effects.

15. Please use a consistent variable name in the job-hopping part. Is HOPPING the same as hopping? Also HOPPING, HOPPING_1, hopping, and HOPPING_2 are not explained in Table 2. Thus, the discussion of the result in this part is still not clear.

Thanks for your helpful comments!

HOPPING is not same as hopping. We took your input and kept the variable names consistent in the job-hopping part, including HOPPING1, HOPPING2, HOPPING3 and HOPPING4. In addition, we have added explanations for these four variables in Table 2, as follows,

Finally, we also rewrite the empirical results in terms of our redefinition of the variables, as follows,

In column (1) to column (4) of Table7, the coefficient for HOPPING1 are -0.106 (p<0.01), -0.028(p<0.05), -0.088(p<0.05) and -0.017 (p<0.1), respectively, which are negative and statistically significant. These findings are consistent with the theoretical prediction that audit offices with a high ethical culture attract auditors that share similar values and beliefs and these auditors have significantly higher audit quality than other auditors.

In column (5) to column (8) of Table7, the coefficient on the explanatory variable HOPPING3 are 0.242 (p<0.05), 0.061(p<0.05), 0.512(p<0.05), and 0.155(p<0.1), which are positive and significant, consistent with the attrition prediction that the audit quality of auditors who were discharged by audit offices is significantly lower than the audit quality of the rest of auditors, and audit offices discharge these auditors who are not compatible with their firm culture.

16. The sub-sample analysis in Table 12 is not clearly explained.

Thanks for your helpful comments! We have rewritten it as your suggestion. It looks better now.

---

## [Decision Letter · Decision Letter 1]

27 Sep 2022

PONE-D-22-12398R1Philanthropy, Audit Firms Culture and Auditor IndependencePLOS ONE

Dear Dr. Wei,

Thank you for submitting your manuscript to PLOS ONE. After careful consideration, we feel that it has merit but does not fully meet PLOS ONE’s publication criteria as it currently stands. Therefore, we invite you to submit a revised version of the manuscript that addresses the points raised during the review process.

We look forward to receiving your revised manuscript.

Kind regards,

Pattanaporn Chatjuthamard, Ph.D.

Academic Editor

PLOS ONE

Journal Requirements:

Reviewers' comments:

Reviewer's Responses to Questions

**Comments to the Author**

1. If the authors have adequately addressed your comments raised in a previous round of review and you feel that this manuscript is now acceptable for publication, you may indicate that here to bypass the “Comments to the Author” section, enter your conflict of interest statement in the “Confidential to Editor” section, and submit your "Accept" recommendation.

Reviewer #1: All comments have been addressed

2. Is the manuscript technically sound, and do the data support the conclusions?

Reviewer #1: Yes

3. Has the statistical analysis been performed appropriately and rigorously? 

Reviewer #1: Yes

4. Have the authors made all data underlying the findings in their manuscript fully available?

Reviewer #1: No

5. Is the manuscript presented in an intelligible fashion and written in standard English?

Reviewer #1: Yes

6. Review Comments to the Author

Reviewer #1: The manuscript's research question appealed to me. It has a well-defined research issue, appropriate methods, and innovative findings. The authors have made some changes based on previous comments. Nevertheless, authors could improve the paper further by rearrange the format of the paper to fit with the PlusOne’s formatting guidelines such as title page. There is still minor typo, missing comma and missing space e.g. abstract “Using 5246 audits in the Chinese market between 2010 and 2012,” page 21 “on decisions made in the audit scenario[61].” Or “accruals and audit fees, than when there are none. [61].”, table 1 “ Financial companies” (i.e. you probably want to write as Less: Financial companies”. ), equation 3 “where ACCRjt is total accruals; TAjt is total asset in year t-1;”.

7. PLOS authors have the option to publish the peer review history of their article (what does this mean?). If published, this will include your full peer review and any attached files.

Reviewer #1: No

---

## [Author Response · Author response to Decision Letter 1]

1 Oct 2022

Response to Reviewer 1 ( PONE-D-22-12398R1)

Reviewer #1: The manuscript's research question appealed to me. It has a well-defined research issue, appropriate methods, and innovative findings. The authors have made some changes based on previous comments. Nevertheless, authors could improve the paper further by rearrange the format of the paper to fit with the PLOS One’s formatting guidelines such as title page. There is still minor typo, missing comma and missing space e.g. abstract “Using 5246 audits in the Chinese market between 2010 and 2012,” page 21 “on decisions made in the audit scenario[61].” Or “accruals and audit fees, than when there are none. [61].”, table 1 “ Financial companies” (i.e. you probably want to write as Less: Financial companies”. ), equation 3 “where ACCRjt is total accruals; TAjt is total asset in year t-1;”.

Thanks for your helpful suggestions!

I correct the details of the minor typo one by one.

(1) According to the PLOS ONE’s formatting guideline, we have further modified the format of the article. It looks better now!

(2) According to the details of the article pointed out by the reviewer, we checked again and revised one by one. For example, Using 5,246 audits in the Chinese market between 2010 and 2012; Less: Financial companies. And we have added all the missing Spaces in the article.

---

## [Decision Letter · Decision Letter 2]

31 Oct 2022

Philanthropy, Audit Firms Culture and Auditor Independence

PONE-D-22-12398R2

Dear Dr. Wei,

We’re pleased to inform you that your manuscript has been judged scientifically suitable for publication and will be formally accepted for publication once it meets all outstanding technical requirements.

Kind regards,

Pattanaporn Chatjuthamard, Ph.D.

Academic Editor

PLOS ONE

Additional Editor Comments (optional):

Reviewers' comments:

Reviewer's Responses to Questions

**Comments to the Author**

1. If the authors have adequately addressed your comments raised in a previous round of review and you feel that this manuscript is now acceptable for publication, you may indicate that here to bypass the “Comments to the Author” section, enter your conflict of interest statement in the “Confidential to Editor” section, and submit your "Accept" recommendation.

Reviewer #1: All comments have been addressed

Reviewer #2: (No Response)

Reviewer #3: All comments have been addressed

2. Is the manuscript technically sound, and do the data support the conclusions?

Reviewer #1: Yes

Reviewer #2: Yes

Reviewer #3: Yes

3. Has the statistical analysis been performed appropriately and rigorously? 

Reviewer #1: Yes

Reviewer #2: Yes

Reviewer #3: Yes

4. Have the authors made all data underlying the findings in their manuscript fully available?

Reviewer #1: No

Reviewer #2: Yes

Reviewer #3: Yes

5. Is the manuscript presented in an intelligible fashion and written in standard English?

Reviewer #1: Yes

Reviewer #2: Yes

Reviewer #3: Yes

6. Review Comments to the Author

Reviewer #1: The authors have tried to address all the previous comments but there are still some typo, missing space and subscription, nevertheless, these mistakes are very few and within acceptable range for published paper.

Reviewer #2: 1. The abstract should include the finding on H3 even thought it was not statistically significant but the fact should be reported.

2. Table 7 result is important and it should be part of the main results of this study as it uses the direct proxy of audit quality which I believe it is a better proxy as compared to other indirect proxies of audit quality such as RE or |DA|.

3. The results on H2 (See Table 5) are not strong as they are significant at 10%. Thus, they should be mentioned on the inference section regarding the 10% significance.

Reviewer #3: (No Response)

7. PLOS authors have the option to publish the peer review history of their article (what does this mean?). If published, this will include your full peer review and any attached files.

Reviewer #1: No

Reviewer #2: No

Reviewer #3: No

---

## [Editor Report · Acceptance letter]

2 Nov 2022

PONE-D-22-12398R2 

Philanthropy, Audit Firms Culture and Auditor Independence 

Dear Dr. Wei:

I'm pleased to inform you that your manuscript has been deemed suitable for publication in PLOS ONE. Congratulations! Your manuscript is now with our production department. 

Kind regards, 

on behalf of

Assoc. Prof. Dr. Pattanaporn Chatjuthamard 

Academic Editor

PLOS ONE